# iVideoGPT: Interactive VideoGPTs are Scalable World Models

**Jialong Wu**[1,*] **Shaofeng Yin**[1,2,*] **Ningya Feng**[1]**, Xu He**[3]**, Dong Li**[3]**, Jianye Hao**[3,4]**,**
**Mingsheng Long**[1,✉]

[1]School of Software, BNRist, Tsinghua University, [2]Zhili College, Tsinghua University
[3]Huawei Noah's Ark Lab, [4]College of Intelligence and Computing, Tianjin University
`wujialong0229@gmail.com, ysf22@mails.tsinghua.edu.cn, mingsheng@tsinghua.edu.cn`

## Abstract

World models empower model-based agents to interactively explore, reason, and plan within imagined environments for real-world decision-making. However, the high demand for interactivity poses challenges in harnessing recent advancements in video generative models for developing world models at scale. This work introduces Interactive VideoGPT (iVideoGPT), a scalable autoregressive transformer framework that integrates multimodal signals—visual observations, actions, and rewards—into a sequence of tokens, facilitating an interactive experience of agents via next-token prediction. iVideoGPT features a novel compressive tokenization technique that efficiently discretizes high-dimensional visual observations. Leveraging its scalable architecture, we are able to pre-train iVideoGPT on millions of human and robotic manipulation trajectories, establishing a versatile foundation that is adaptable to serve as interactive world models for a wide range of downstream tasks. These include action-conditioned video prediction, visual planning, and model-based reinforcement learning, where iVideoGPT achieves competitive performance compared with state-of-the-art methods. Our work advances the development of interactive general world models, bridging the gap between generative video models and practical model-based reinforcement learning applications. Code and pre-trained models are available at `https://thuml.github.io/iVideoGPT`.

## 1 Introduction

Recent years have witnessed remarkable advancements in generative models of multimodal contents, including text [1], audio [9], and images [22], with video generation now emerging as a new frontier [11]. A particularly significant application of these generative video models, learned in an unsupervised way on diverse Internet-scale data, is to construct predictive world models [53, 28] at scale. These world models are expected to accumulate commonsense knowledge about how the world works, enabling the prediction of potential future outcomes (e.g., visual observations and reward signals) based on the actions of agents. By leveraging these world models, agents employing model-based reinforcement learning (RL) can imagine, reason, and plan inside world models [20, 29], thus acquiring new skills more safely and efficiently with a handful of trials in the real world.

Despite the fundamental connection, significant gaps remain between generative models for video generation and visual world models for agent learning. One primary challenge is achieving the best of both interactivity and scalability. In model-based RL, world models predominantly utilize recurrent network architecture. This design naturally allows the transition of observations or latent states conditioned on actions in each step, facilitating interactive behavior learning [29, 80, 34]. However, these recurrent models mostly focus on games or simulated environments with simple

---

[*]Equal Contribution

38th Conference on Neural Information Processing Systems (NeurIPS 2024).

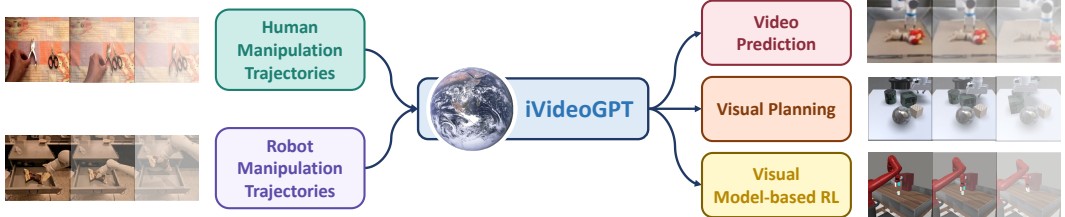

Figure 1: Practical applications of iVideoGPT, which is designed to scale, allowing pre-training on millions of human and robotic manipulation trajectories. This results in a single, versatile foundation of interactive world models, adaptable to a wide range of downstream tasks.

visuals and have limited capability to model complex, in-the-wild data at scale [48, 81]. On the other hand, Internet-scale video generative models [37, 7, 11] can synthesize realistic long videos that are controllable via text descriptions [109] or future action sequences [101] at the beginning of generation. Although suitable for high-level planning [19], their trajectory-level interactivity does not provide sufficient granularity needed by agents to intervene step-by-step during the simulation to learn precise basic skills efficiently. This dilemma naturally raises the question:

*How can we leverage the advancements in scalable video generative models for developing interactive visual world models?*

In this work, we explore world models that are both interactive and scalable within a GPT-like autoregressive transformer framework [90, 75]. Pioneering efforts have been made recently through diffusion models [102] and masked generative models [12]. Nevertheless, utilizing autoregressive transformers offers distinct advantages such as seamless integration with the established Large Language Model (LLM) ecosystem [110] and greater flexibility in handling diverse conditions without the need for specific architectural modifications like adapter modules [77, 107]. We present *Interactive VideoGPT (iVideoGPT)*, a scalable world model architecture that incorporates multimodal signals, including visual observations, actions, and rewards, in an interactively autoregressive manner. Unlike multimodal LLMs that discretize visual observations into tokens frame-by-frame using image tokenizers [55], a key innovation of iVideoGPT for enhancing scalability is to learn compressive tokenization for each observation conditioned on rich contextual observations, achieving an asymptotic $16\times$ reduction in token sequence length. We highlight that more compact video tokenization could not only facilitate more efficient training and generation but also enhance video quality. This is achieved by decoupling context from dynamics, allowing the model to focus on predicting the motion of objects while maintaining temporal consistency within the scene [99].

We demonstrate a series of practical applications of iVideoGPT for visual robotic manipulation, as illustrated in Figure 1. Mirroring the two-phase approach popularized by LLMs, our method involves pre-training followed by domain-specific adaptation. During pre-training, iVideoGPT is scalable for action-free video prediction across a mixture of over one million robotic and human manipulation trajectories [70, 25]. The pre-trained iVideoGPT serves as a single, adaptable foundation of interactive world models for various downstream tasks, such as action-conditioned video prediction [21, 16], visual planning [86], and visual model-based RL [105]. Additionally, we showcase the pre-trained transformer's preliminary zero-shot video generation capability without fine-tuning, requiring only tokenizer adaptation for unseen domains. We further explore a variant of iVideoGPT for goal-conditioned video prediction, underscoring the flexibility of sequence modeling.

The main contributions of this work can be summarized as follows:

- We introduce Interactive VideoGPT (iVideoGPT), an autoregressive transformer architecture for scalable world models, which features compressive tokenization for visual observations.
- We pre-train iVideoGPT on a large-scale dataset comprising millions of robotic and human manipulation trajectories and adapt it to domain-specific tasks. The pre-trained models have been publicly available to encourage further research.
- Extensive experiments covering video prediction, visual planning, and visual model-based RL demonstrate that iVideoGPT can simulate accurate and realistic experiences and provide competitive performance compared with state-of-the-art methods.

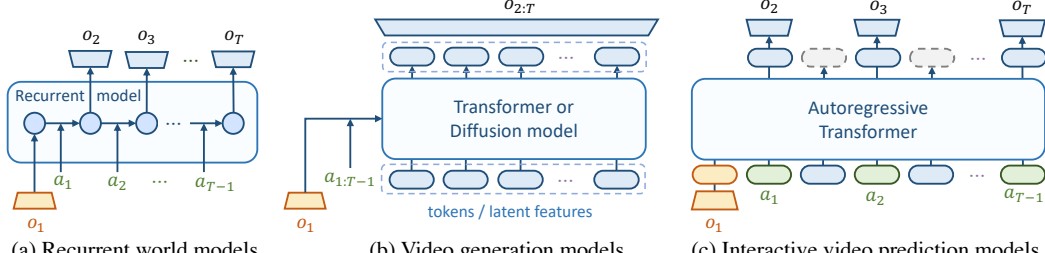

Figure 2: Conceptual comparison among architectures, illustrated using a single context frame ($T_0 = 1$) for simplicity. (a) Recurrent architectures for world models like Dreamer [29] and MuZero [80] provide step-level interactivity but limited scalability. (b) Recent video generation advancements like VideoGPT [101] and Stable Video Diffusion [8, 7] use non-causal temporal modules that can only offer trajectory-level interactivity. (c) Our model utilizes an autoregressive transformer that separately maps each step into a sequence of tokens, achieving both scalability and interactivity.

## 2   Problem Formulation

A world model is an internal model learned by the agent to simulate the environment. This environment is typically modeled as a *partially observable Markov decision process (POMDP)* $(\mathcal{S}, \mathcal{O}, \phi, \mathcal{A}, p, r, \gamma)$. At each step, $s_t \in \mathcal{S}$ represents the underlying state of the environment, and $o_t = \phi(s_t)$ is the observation received by the agent, only providing incomplete information of $s_t$. After taking an action $a_t \in \mathcal{A}$, $p(s_{t+1}|s_t, a_t)$ defines the transition probability from state $s_t$ to $s_{t+1}$. The agent also receives immediate rewards $r_{t+1} = r(s_t, a_t)$, and aim to learn a policy $\pi$ such that $a_t \sim \pi(o_{1:t})$ maximizing the $\gamma$-discounted accumulated rewards $\mathbb{E}_{p,\pi}[\sum_t \gamma^{t-1} r_t]$.

While world models can be learned from many types of data, video is one modality that is task-agnostic, widely available, and embeds broad knowledge that can be learned in a self-supervised way. Thus, we formulate learning world models for visual control as an *interactive video prediction* problem [102, 12] where $\mathcal{O} = \mathbb{R}^{H \times W \times 3}$ is the space of video frames[2]. Concretely, given a short history visual observations of $T_0$ frames $o_{1:T_0}$, at each step $t = T_0, \ldots, T-1$, the agent takes an action $a_t$ based on its policy and previous imagined observations, and then the world model need to approximate and sample the transition $p(o_{t+1}, r_{t+1} \mid o_{1:t}, a_{T_0:t})$ to feedback the agent.

As depicted in Figure 2, a majority of advanced video generation models [101, 8, 104], including VideoGPT, can not deal with the interactive video prediction problem because they design non-causal modules fusing information along the temporal dimension, lacking the ability for causal, intermediate action control during generation (see extended discussion in Appendix C.2). Existing world models in the literature of MBRL [29, 80], such as Dreamer, utilize recurrent architecture but lack scalability.

## 3   Interactive VideoGPT

In this section, we introduce Interactive VideoGPT, a scalable world model architecture with great flexibility to integrate multimodal signals, including visual observations, actions, rewards, and other potential sensory inputs. At its core, iVideoGPT consists of a compressive tokenizer to discretize video frames and an autoregressive transformer predicting subsequent tokens (Section 3.1). This model can acquire common knowledge of motions and interactions in various scenes through pre-training on diverse human and robotic manipulation videos (Section 3.2) and then effectively transfer to downstream tasks incorporating additional modalities (Section 3.3).

### 3.1   Architecture

**Compressive tokenization.**   Transformers particularly excel in operating over sequences of discrete tokens. VQGAN [22] is a commonly used visual tokenizer that converts from raw pixels to discrete tokens. Instead of using an image tokenizer to discretize each frame independently [55, 63, 27], leading to rapidly increasing sequence lengths, or using a 3D tokenizer that compresses videos

---

[2]Due to this connection, we use the terms "video frame" and "visual observation" interchangeably.

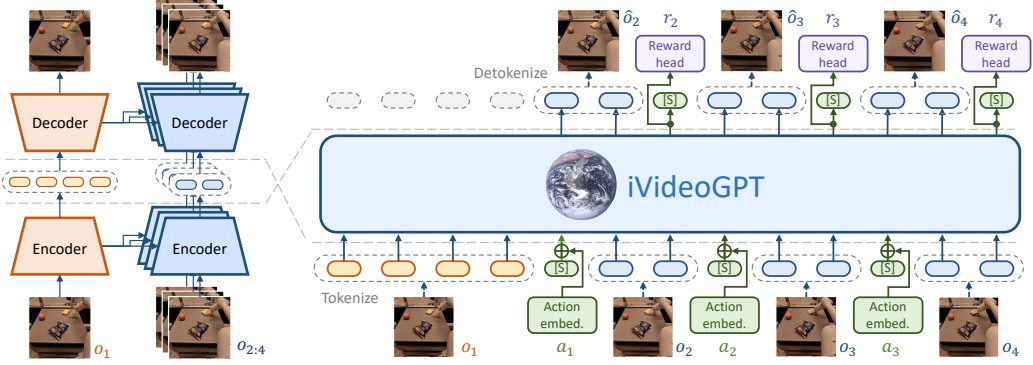

(a) Compressive tokenization      (b) Interactive prediction with Transformers

Figure 3: Architecture of iVideoGPT, simplified to show only a single context frame ($T_0 = 1$). (a) Compressive tokenization utilizes a conditional VQGAN that discretizes future frames conditioned on context frames to handle temporal redundancy, significantly reducing the number of video tokens. (b) An autoregressive transformer integrates multimodal signals—visual observations, actions, and rewards—into a sequence of tokens, enabling interactive agent experiences through next-token prediction. Actions and rewards are optional and not included in action-free video pre-training.

spatiotemporally at the expense of interactivity [101, 104], we propose to tokenize videos with a novel conditional VQGAN consisting of dual encoders and decoders $\{(E_c, D_c), (E_p, D_p)\}$. As illustrated in Figure 3a, initial context frames $o_{1:T_0}$, rich in contextual information, are independently tokenized and reconstructed through $N$ tokens: $z_t^{(1:N)} = E_c(o_t), \hat{o}_t = D_c(z_t)$ for $t = 1, \ldots, T_0$. In contrast, due to the temporal redundancy between context and future frames, only essential dynamics information, such as the position and pose of moving objects, needs to be encoded. This is achieved using a conditional encoder and decoder, which require a far smaller number of $n$ tokens ($n \ll N$):

$$z_t^{(1:n)} = E_p(o_t|o_{1:T_0}), \hat{o}_t = D_p(z_t|o_{1:T_0}) \quad \text{for } t = T_0 + 1, \ldots, T. \tag{1}$$

We implement this conditioning mechanism using cross-attention between multi-scale feature maps (see details in Appendix A.1). Overall, the proposed tokenizer is trained with the following objective:

$$\mathcal{L}_{\text{tokenizer}} = \sum_{t=1}^{T_0} \mathcal{L}_{\text{VQGAN}}(o_t; E_c(\cdot), D_c(\cdot)) + \sum_{t=T_0+1}^{T} \mathcal{L}_{\text{VQGAN}}(o_t; E_p(\cdot|o_{1:T_0}), D_p(\cdot|o_{1:T_0})), \tag{2}$$

where $\mathcal{L}_{\text{VQGAN}}(o; E, D)$ is a combination of a $L_1$ reconstruction loss, a commitment loss [89], a perceptual loss [44], and optionally an adversarial loss [22].

There are primarily two benefits of the proposed tokenization. First, it significantly reduces the sequence length of tokenized videos, which grows linearly with the number of frames but at a much smaller rate $n$. In this work, we set $N = 16 \times 16$ and $n = 4 \times 4$, resulting in an asymptotic reduction of $16\times$, facilitating faster rollouts for model-based planning and reinforcement learning. Second, by conditional encoding, transformers predicting subsequent tokens can maintain temporal consistency of the context much easier and focus on modeling essential dynamics information [99]. We discuss the assumptions and limitations of our tokenization in Section 6.

**Interactive prediction with Transformers.** After tokenization, the video is flattened into a sequence of tokens: $x = (z_1^{(1)}, \ldots, z_1^{(N)}, [\texttt{S}], z_2^{(1)}, \ldots, z_2^{(N)}, \ldots, [\texttt{S}], z_{T_0+1}^{(1)}, \ldots, z_{T_0+1}^{(n)}, \ldots)$ with a length of $L = (N+1)T_0 + (n+1)(T - T_0) - 1$. Special slot tokens $[\texttt{S}]$ are inserted to delineate frame boundaries and facilitate the integration of extra low-dimensional modalities such as actions (see Section 3.3 for details). As Figure 3b, a GPT-like autoregressive transformer is utilized for interactive video prediction through next-token generation frame-by-frame. In this work, we take the model size of GPT-2 [76] but adopt the LLaMA architecture [87] in order to embrace the latest innovations for LLM architecture, applying pre-normalization using RMSNorm [106], SwiGLU activation function [83], and rotary positional embeddings [85].

## 3.2 Pre-Training

Large language models can gain extensive knowledge from Internet text in a self-supervised way via next-word prediction. Similarly, the *action-free video pre-training* paradigm for world models [81, 99, 62] involves video prediction as a pre-training objective, providing Internet-scale supervision with physical world knowledge absent in LLMs. We pre-train iVideoGPT on this generic objective, applying a cross-entropy loss to predict subsequent video tokens:

$$\mathcal{L}_{\text{pre-train}} = -\sum\nolimits_{i=(N+1)T_0+1}^{L} \log p(x_i|x_{<i}), \tag{3}$$

where $L$ is the total sequence length and $(N+1)T_0+1$ marks the first token index of the frames to be predicted. Notably, we do not train iVideoGPT to generate context frames, making its capacity focus on dynamics information, as previously discussed.

**Pre-training data.**  While there are numerous videos available on the Internet, due to computational limitations, we specifically pre-train iVideoGPT for the robotic manipulation domain. We leverage a mixture of 35 datasets from the Open X-Embodiment (OXE) dataset [70] and the Something-Something v2 (SSv2) dataset [25], totaling 1.4 million trajectories (see Appendix A.2 for details). OXE is a diverse collection of robot learning datasets from a variety of robot embodiments, scenes, and tasks. These datasets are highly heterogeneous but can be easily unified in the action-free video prediction task. To further enhance the diversity, we also include SSv2, a dataset of human-object interaction videos, as previous work has demonstrated knowledge transfer from these human manipulation videos for learning a world model for robotic manipulation tasks [99, 62].

**Flexibility of sequence modeling.**  A sequence of tokens provides a flexible way to specify tasks, inputs, and outputs [60, 76]. To preliminarily showcase this flexibility, we introduce a variant of iVideoGPT for goal-conditioned video prediction: $p(o_{T_0+1:T}|o_{1:T_0}, o_T)$, where the model predicts a video sequence reaching a specified goal observation $o_T$. This is simply achieved by rearranging the frame sequence as $\tilde{o}_{1:T} = (o_T, o_1, o_2, \ldots, o_{T-1})$ while keeping the architecture and training procedure consistent as above (see details in Appendix A.2). Qualitative results of goal-conditioned prediction are shown in Figure 4, with further exploration left for future work[3].

## 3.3 Fine-Tuning

**Action conditioning & reward prediction.**  Our architecture is also designed to flexibly incorporate additional modalities for learning interactive world models, as illustrated in Figure 3b. Actions are integrated by linear projection and adding to the slot token embeddings. For reward prediction, instead of learning independent reward predictors, we add a linear head to the last token's hidden state of each observation. This multi-task learning approach can enhance the model's focus on task-relevant information, thereby improving prediction accuracy for control tasks [57]. We use a mean-squared error loss for reward prediction in addition to the cross-entropy loss in Eq. (3).

**Tokenizer adaptation.**  We choose to update the full model, including the tokenizer, for downstream tasks, finding this strategy more effective than parameter-efficient fine-tuning methods [39]. This is likely due to the limited diversity of our pre-trained data compared to Internet-scale images, which, while extensive, may also not adequately cover specific real-world applications like robotics. Minimal literature explores adapting a VQGAN tokenizer to domain-specific data. As our tokenization is designed for decoupling dynamics information from context conditions, we hypothesize that while our model may encounter unseen objects like different robot types in downstream tasks, the fundamental knowledge of physics—such as motions and interactions—learned by the transformer from diverse scenes is commonly shared. This hypothesis is supported by our experiments transferring iVideoGPT from mixed pre-training data to the unseen BAIR dataset [21], where the pre-trained transformer can zero-shot generalize to predict natural motions, requiring only the tokenizer to be fine-tuned for unseen robot grippers (see Figure 8). This property is particularly important for scaling GPT-like transformers to large sizes, enabling lightweight alignment across domains while keeping the transformer intact. We leave an in-depth analysis of tokenizer adaptation for future work.

---

[3]Unless otherwise specified, action- and goal-free video prediction is used as the default pre-training objective to obtain pre-trained models for all experiments.

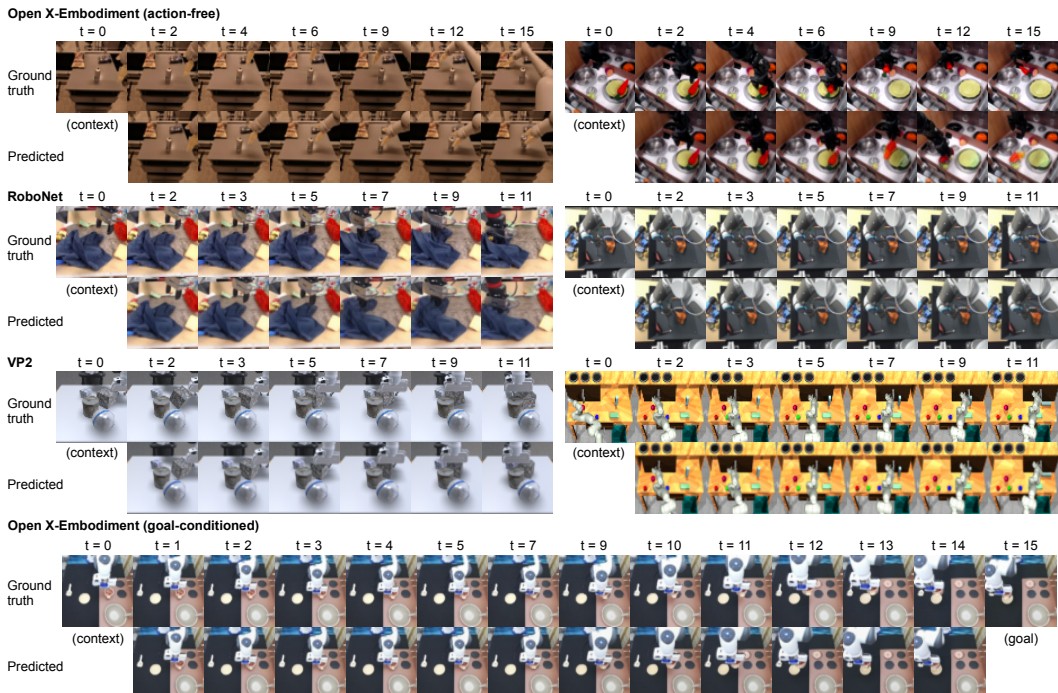

Figure 4: Qualitative evaluation: video prediction results of iVideoGPT on Open X-Embodiment, RoboNet, and VP$^2$. Zoom in for details. Extended examples can be found in Appendix B.1.

## 4 Experiments

In this section, we evaluate iVideoGPT in three different control-relevant settings and compare its performance with prior state-of-the-art methods. We demonstrate that iVideoGPT is versatile to provide competitive performance across a range of tasks (Section 4.1, 4.2, and 4.3) and conduct in-depth analysis to understand the tokenization and prediction ability, data efficiency, model scaling, and computational efficiency (Section 4.4). Experimental details can be found in Appendix A.

### 4.1 Video Prediction

**Setup.** The BAIR robot pushing dataset [21] consists of 43k training and 256 test videos, where we predict 15 frames from a single initial frame, a standard protocol of prior works. The RoboNet dataset [16] contains 162k videos across 7 robotic arms. Following prior works, we use 256 videos for testing, predicting 10 frames from two frames. Notably, RoboNet overlaps with our pre-training data OXE, from which we have carefully filtered test videos. We compare against a variety of video prediction models, including variational [91, 98, 4], diffusion [93], masked [104, 27], and autoregressive models [101], across four metrics: FVD [88], PSNR [40], SSIM [97], and LPIPS [108].

**Results.** As shown in Table 1, iVideoGPT provides competitive performance compared to state-of-the-art methods, MAGVIT [104] for BAIR and FitVid [4] for RoboNet, while achieving both interactivity and scalability in its architecture. Initially pre-trained action-free, our model flexibly allows for action-conditioning, which notably improves FVD for BAIR by almost 20%. Although primary experiments are at a low resolution of $64 \times 64$, iVideoGPT can be easily extended to $256 \times 256$ for RoboNet. We highlight that MaskViT, a prior method leveraging per-frame tokenization, suffers from temporal inconsistency and flicker artifacts in VQGAN reconstructions. Our model, which employs compressive tokenization conditioned on consistent contextual information, improves this and significantly outperforms MaskViT. For qualitative results, refer to Figure 4.

### 4.2 Visual Planning

**Setup.** VP$^2$ is a control-centric benchmark [86] that evaluates video prediction models for visual model-predictive control (MPC) [24, 20] across four Robosuite [117] and seven RoboDesk tasks [47].

Table 1: Video prediction results on the BAIR robot pushing and RoboNet datasets. We report the mean and standard deviation for each metric calculated over three runs. "-" marks that the value is not reported in the original papers. LPIPS and SSIM scores are scaled by 100 for convenient display.

| **BAIR** [21] | FVD↓ | PSNR↑ | SSIM↑ | LPIPS↓ |
|---|---|---|---|---|
| *action-free & 64×64 resolution* | | | | |
| VideoGPT [101] | 103.3 | - | - | - |
| MaskViT [27] | 93.7 | - | - | - |
| FitVid [4] | 93.6 | - | - | - |
| MCVD [93] | 89.5 | 16.9 | 78.0 | - |
| MAGVIT [104] | **62.0** | 19.3 | 78.7 | 12.3 |
| iVideoGPT (ours) | 75.0±0.20 | **20.4**±0.01 | **82.3**±0.05 | **9.5**±0.01 |
| *action-conditioned & 64×64 resolution* | | | | |
| MaskViT [27] | 70.5 | - | - | - |
| iVideoGPT (ours) | **60.8**±0.08 | **24.5**±0.01 | **90.2**±0.03 | **5.0**±0.01 |

| **RoboNet** [16] | FVD↓ | PSNR↑ | SSIM↑ | LPIPS↓ |
|---|---|---|---|---|
| *action-conditioned & 64×64 resolution* | | | | |
| MaskViT [27] | 133.5 | 23.2 | 80.5 | 4.2 |
| SVG [91] | 123.2 | 23.9 | 87.8 | 6.0 |
| GHVAE [98] | 95.2 | 24.7 | 89.1 | 3.6 |
| FitVid [4] | **62.5** | **28.2** | 89.3 | **2.4** |
| iVideoGPT (ours) | 63.2±0.01 | 27.8±0.01 | **90.6**±0.02 | 4.9±0.00 |
| *action-conditioned & 256×256 resolution* | | | | |
| MaskViT [27] | 211.7 | 20.4 | 67.1 | 17.0 |
| iVideoGPT (ours) | **197.9**±0.66 | **23.8**±0.00 | **80.8**±0.01 | **14.7**±0.01 |

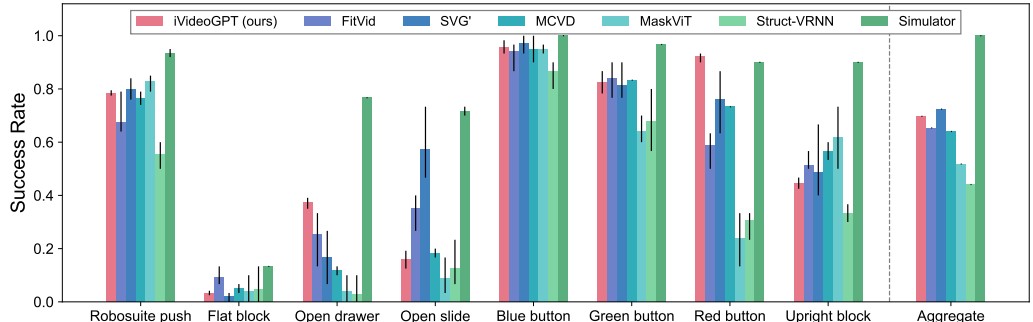

Figure 5: Visual MPC results on the VP$^2$ benchmark. We report the mean and min/max performance of iVideoGPT over 3 control runs. On the right, we show the mean scores averaged across all tasks except flat block due to low simulator performance, normalized by the performance of the simulator.

Each environment's training dataset includes noisy scripted interaction trajectories. Following the protocol from the original benchmark paper, we trained iVideoGPT on 5k trajectories for Robosuite and 35k for RoboDesk, comparing our models with established baselines.

**Results.** Figure 5 presents the success rates of iVideoGPT compared to baseline models. While Tian et al. [86] observed that excellent perceptual metrics do not always correlate with effective control performance, iVideoGPT outperforms all baselines in two RoboDesk tasks with a large margin and achieves comparable average performance to the strongest model, SVG′ [91]. In Appendix C.3, we analyze iVideoGPT's suboptimal performance on the open slide task, which is attributed to both limitations of discretization in our model and imperfect built-in reward design of the benchmark.

### 4.3 Visual Model-based Reinforcement Learning

**Setup.** We conduct experiments on six robotic manipulation tasks of varying difficulty from Meta-World [105]. Leveraging iVideoGPT as interactive world models, we have developed a model-based RL method adapted from MBPO [42], which augments the replay buffer with synthetic rollouts to train a standard actor-critic RL algorithm (see Appendix A.5 for the pseudo-code). Our implementation builds upon DrQ-v2 [103], a state-of-the-art visual model-free RL method. We also compare against a state-of-the-art model-based RL algorithm, DreamerV3 [32], with and without world model pre-training [81].

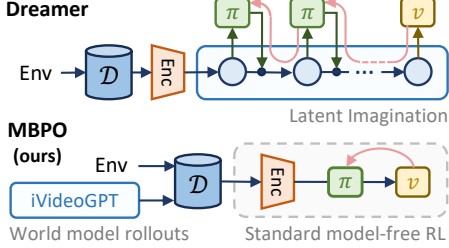

Figure 6: Powerful iVideoGPTs enable a simple yet performant MBRL algorithm, decoupling model rollouts and policy learning.

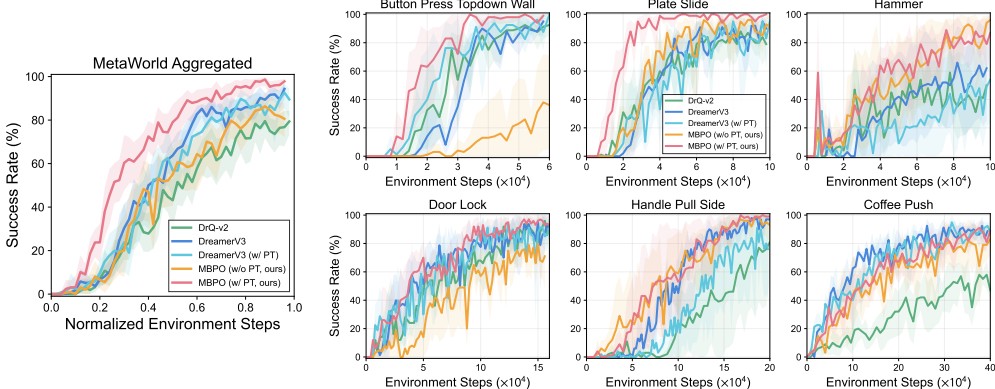

Figure 7: Visual model-based RL on Meta-world. *(Left)* Aggregated results report interquartile mean and 95% confidence interval (CI) [2] across a total of 30 runs over six tasks. *(Right)* Individual results for each task, report mean and 95% CI across five runs, measuring success rates over 20 evaluation episodes. *PT* denotes pre-training.

**Results.** Figure 7 shows that our model-based algorithm not only remarkably improves the sample efficiency over its model-free counterpart but also matches or exceeds the performance of DreamerV3. To our knowledge, this reports the *first successful application of MBPO to visual continuous control tasks*. These results highlight the opportunity, with powerful world models, to eliminate the need for latent imagination—a common strategy used in advanced MBRL systems to train policies on rollouts of latent states within world models [29, 80] (see comparison in Figure 6). Our development of performant MBRL algorithms decouples model and policy learning, where iVideoGPT simply serves as a drop-in replacement of the environment. This can substantially simplify the design space, thereby greatly enhancing the practicality and effectiveness of MBRL algorithms in real-world applications.

**Comparison to recurrent world models.** We argue that recurrent world models lack the capacity for large-scale pre-training on real-world data—a crucial capability for modern foundation models. To validate this, we pre-train DreamerV3 XL (200M parameters, comparable to iVideoGPT) on the same dataset. As shown in Figure 11 in the Appendix, DreamerV3 fails to capture natural robot dynamics, yielding low-quality, blurred predictions. Further evaluation on the Meta-World benchmark in Figure 7 reveals that DreamerV3 cannot benefit from such ineffective pre-training.

### 4.4 Model Analysis

**Zero-shot prediction.** We first analyze the zero-shot video prediction ability of large-scale pre-trained iVideoGPT on the unseen BAIR dataset. Interestingly, we observe in the second row of Figure 8 that iVideoGPT, without fine-tuning, predicts a natural movement of a robot gripper—albeit a different one from our pre-training dataset. This indicates that while, due to insufficient diversity of pre-training data, our model has a limited ability of zero-shot generalization to completely unseen robots, it effectively separates scene context from motion dynamics. In contrast, with an adapted tokenizer, the transformer that is not fine-tuned itself successfully transfers the pre-trained knowledge and predicts movements for the new robot type in the third row, providing a similar perceptual quality as the fully fine-tuned transformer in the fourth row. Quantitative results can be found in Figure 9a.

**Few-shot adaptation.** Large-scale pre-trained models have proven effective, especially in data-scarce scenarios. Figure 9a shows iVideoGPT's performance when fine-tuned with various sizes of action-free BAIR trajectories. We observe that pre-training offers minimal benefits when full downstream data is available, yet the advantages become significant under data scarcity (with 100 or 1,000 trajectories). We also adapt iVideoGPT using 1,000 action-conditioned BAIR trajectories, achieving an FVD of 82.3. The fast adaptation ability with a handful of data is particularly crucial in model-based RL. As shown in Figure 7, world models trained from scratch may generate inaccurate predictions, thereby degenerating the sample efficiency that is vital for model-based agents.

**Model scaling.** All previous experiments are conducted using an iVideoGPT with 12 transformer layers and 768-dimensional hidden states (138M parameters). To initially investigate the scaling

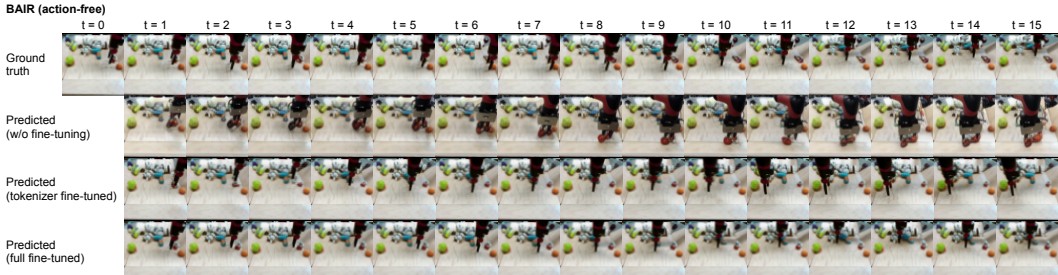

Figure 8: Zero-shot prediction by pre-trained transformer in iVideoGPT. Without fine-tuning, the transformer predicts natural movements of a different robot gripper using the pre-trained tokenizer *(second row)* but accurately predicts for the correct gripper type with an adapted tokenizer *(third row)*.

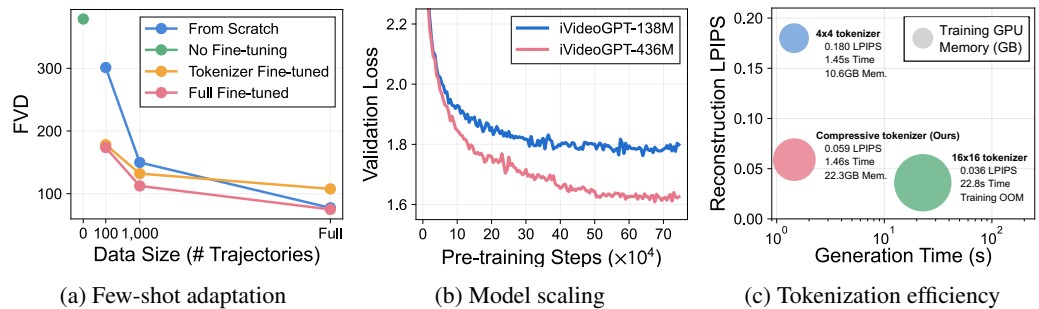

(a) Few-shot adaptation      (b) Model scaling      (c) Tokenization efficiency

Figure 9: Model analysis. (a) Video prediction results with various fine-tuning strategies and data sizes on BAIR. (b) Validation losses for the 138M and 436M transformer models on the pre-training dataset. (c) Computational efficiency and reconstruction quality of different tokenizers.

behavior of our model, we trained a larger iVideoGPT with 24 layers and 1024-dimensional hidden states (436M parameters). Figure 9b illustrates the validation loss curves on the pre-trained dataset. It shows that (1) the validation loss (perplexity) continues to decrease regardless of model size, and (2) increasing the model size accelerates the loss decrease. These results align with our expectation that larger model sizes and increased computation [48] can build more powerful iVideoGPTs.

**Tokenization efficiency.** We evaluate the effectiveness of our compressive tokenization by comparing it against standard VQGAN tokenizers that independently convert each frame into $16 \times 16$ and $4 \times 4$ tokens. We train three tokenizers from scratch on RoboNet for the same number of steps. As Figure 9c, the tokenizer with $4 \times 4$ tokens suffers from low reconstruction quality due to its insufficient capacity. Our proposed tokenization method slightly compromises reconstruction quality compared to the standard $16 \times 16$ tokenizer but can provide more consistent contextual information, which is beneficial for video prediction tasks. More importantly, it significantly enhances computational efficiency with a significantly fewer amount of tokens, which greatly saves time and memory, allowing us to scale the model size with fewer costs (see quantitative results in Appendix B.5).

**Context-dynamics decoupling.** Our tokenizer is designed with a bottleneck of much fewer tokens, focusing only on capturing necessary dynamics information for future frames while sharing contextual information with initial frames to reconstruct raw pixels. To explicitly visualize this decoupling of context and dynamics information, we drop cross-attention blocks to context frames in the decoder when reconstructing future frames. The results in Figure 10 show that the decoder can still reproduce the movement trajectories accurately but with minimal contextual information. This visualization supports the explanation of our model's generalization capability shown in Figure 8.

**Goal-conditioned prediction.** In Figure 4, we also showcase video prediction generated by goal-conditioned iVideoGPT, pre-trained on massive human and robotic videos (Section 3.2). Unlike action-free prediction, which often results in trajectories diverging from the ground truth, the goal-conditioned model produces more accurate paths to reach specified goals. We believe this highlights the potential of leveraging the flexibility of a unified sequence modeling paradigm.

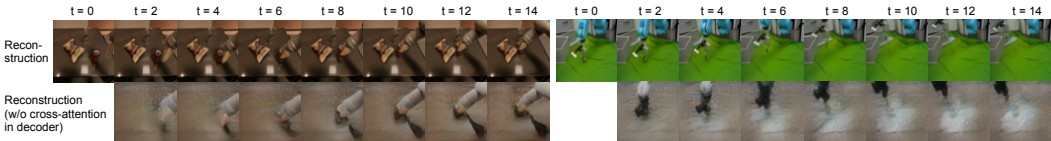

Figure 10: Context-dynamics decoupling in our compressive tokenization. By removing cross-attention from future frames to context frames, the decoder can still reconstruct a trajectory that moves in the same way as the original, but the visual context is almost entirely missing.

## 5 Related Work

**World models for visual control.** Learning general world models in visual domains remains a significant challenge in model-based reinforcement learning. A straightforward method involves learning action-conditioned video prediction models [69, 45]. Advanced model-based RL algorithms [29, 31, 32, 80, 34, 33] utilize latent imagination for more efficient and accurate rollouts but complicate themselves by tightly coupling model and policy learning. We show that this complexity can be reduced with powerful world models that have accumulated generalizable knowledge beyond specific tasks. Recent efforts to facilitate this include leveraging scalable architectures like transformers [63] and pre-training from large-scale data [99, 62]. Of particular relevance to our work are UniSim [102] and Genie [12], which have developed extensively trained world models with diffusion and masked models, respectively, though neither is publicly available. Our work distinguishes itself by utilizing a generic autoregressive transformer framework, advancing the flexibility of scalable world models.

**Video generation and prediction.** Recent developments in Internet-scale video generation models now enable the synthesis of realistic videos conditioned on class labels, text descriptions, and initial frames—the last one also known as the video prediction problem. Various models have been developed, including deterministic RNNs [84, 96], variational autoencoders [18, 3, 30, 4], diffusion [38, 11], masked [104, 27], and autoregressive models [101, 50, 55]. However, most recent works do not treat video prediction as a dynamics modeling problem and perform spatiotemporal compression [101, 8], thus providing limited interactivity to serve as world models. We achieve both compressive tokenization and interactivity by context-aware representation, employing cross-attention mechanisms with minimal inductive bias. This method diverges from previous techniques that rely on motion vectors [43] or optical flows [52] and offers a more generic form of video tokenization.

## 6 Discussion

We introduced Interactive VideoGPT (iVideoGPT), a generic and efficient world model architecture that leverages a scalable autoregressive transformer to integrate multimodal signals into a sequence of tokens, providing an interactive agent experience via next-token prediction. iVideoGPT has been pre-trained on millions of human and robotic manipulation trajectories and adapted to a wide range of downstream tasks. As a powerful foundation of world models, it enables accurate and generalizable video prediction as well as simplified yet performant model-based planning or reinforcement learning.

**Limitations and future work.** While iVideoGPT marks significant progress, there is substantial room for improvement. We found limited diversity in publicly available robotic data, including the large-scale Open X-Embodiment dataset, and initiated efforts to transfer knowledge from human videos [25]. We believe iVideoGPT should be pre-trained on more extensive data [26] to bridge knowledge between humans and robots. This also requires iVideoGPT to incorporate more modalities, such as multi-view observations, proprioceptive robot states, and actions, within the unified formulation beyond action-free video prediction. Specifically, to process high-dimensional visual observations, our compressive tokenization assumes that initial frames provide sufficient contexts for future frames, which works for low-level control tasks as model-based agents often foresee tens of steps, but may falter in scenarios with long videos and significant camera motion. This issue can be mitigated by keyframe extraction [51] but leaves an important future avenue of exploration. Finally, extending to more complex real-robot tasks is essential, as the benefits of model scaling to even larger sizes remain unobserved in this work within visually simple simulation for downstream control tasks.

## Acknowledgements

We would like to thank many colleagues, in particular Yuchen Zhang, Lanxiang Xing, and Haixu Wu, for their valuable discussion. This work was supported by the National Natural Science Foundation of China (U2342217 and 62021002), the BNRist Project, the Huawei Innovation Fund, and the National Engineering Research Center for Big Data Software.

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

# A  Implementation and Experimental Details

The main hyperparameters of our experiment are detailed in Tables 2, 3, and 5. In this section, we provide a comprehensive explanation of all experimental details.

## A.1  Architecture

Table 2: Hyperparameters of iVideoGPT architectures.

| VQGAN | Low-resolution | High-resolution |
|---|---|---|
| Parameters | 114M | 310M |
| Resolution | $64 \times 64$ | $256 \times 256$ |
| Down blocks | 3 | 5 |
| Down layers per block | 2 | 2 |
| Down channels | [128, 256, 512] | [128, 256, 256, 512, 768] |
| Mid block attention | False | False |
| Up blocks | 3 | 5 |
| Up layers per block | 3 | 3 |
| Up channels | [512, 256, 128] | [768, 512, 256, 256, 128] |
| Embedding dim | 64 | 64 |
| Codebook size | 8192 | 8192 |
| Norm | GroupNorm | GroupNorm |
| Norm group | 32 | 32 |
| Activation | SiLU | SiLU |
| Max cross-att. resolution | 16 | 32 |
| **Transformer** | **Small** | **Medium** |
| Parameters | 138M | 436M |
| Layers | 12 | 24 |
| Heads | 12 | 16 |
| Hidden dim | 768 | 1024 |
| Feedforward dim | 3072 | 4096 |
| Dropout | 0.1 | 0.1 |
| Activation | SiLU | SiLU |

**Tokenizier.**  As illustrated in Figure 3, we use a conditional VQGAN for compressive tokenization. This comprises two encoder-decoder pairs: $(E_c, D_c)$ for context frames (referred to as the *context encoder-decoder*) and $(E_p, D_p)$ for future frames (referred to as the *prediction encoder-decoder*). Both pairs share the same architecture (detailed in Table 2), but the prediction encoder-decoder has a tighter bottleneck, focusing solely on encoding dynamic information. Specifically, it uses a $4 \times 4$ convolution to downsample $16 \times 16$ embeddings into $4 \times 4$ before looking up the codebook. Consequently, the prediction encoder-decoder needs to be conditioned on the features of the context encoder-decoder to incorporate rich contextual information. This conditioning is implemented via a multi-scale cross-attention mechanism, similar to ContextWM [99].

The intuition behind the multi-scale cross-attention across feature maps is as follows: the context encoder extracts contextual features at varying levels of abstraction, while the prediction encoder uses cross-attention to adaptively filter out contextual information and distill dynamics information. During decoding, the prediction decoder blocks employ cross-attention to retrieve contextual information at corresponding levels, facilitating the gradual reconstruction of the scene. This framework enhances the model's ability to understand and manipulate complex scenes by focusing on dynamic changes, rather than being overwhelmed by irrelevant visual details.

Specifically, at the end of each encoder block, let $F_c^l \in \mathbb{R}^{c \times h \times w}$ be the feature map of a context frame, and $F_p^l \in \mathbb{R}^{c \times h \times w}$ be the feature map of a future frame. Before being processed by the next

Table 3: Hyperparameters of iVideoGPT training and evaluation.

| VQGAN | Low-resolution $(64 \times 64)$ | | | | High-resolution $(256 \times 256)$ | |
|---|---|---|---|---|---|---|
| | Pre-train | BAIR | RoboNet | VP$^2$ | Pre-train | RoboNet |
| GPU days | 17 | 2 | 8 | 4 | 16 | 9 |
| Training steps | $1 \times 10^6$ | $2 \times 10^5$ | $6 \times 10^5$ | $2 \times 10^5$ | $2.5 \times 10^5$ | $1.5 \times 10^5$ |
| Disc. start | - | - | - | - | - | $1 \times 10^5$ |
| Batch size | 64 | 64 | 64 | 64 | 32 | 32 |
| Sequence length | 16 | 16 | 12 | 12 | 16 | 12 |
| Context frames | 2 | 1 | 2 | 2 | 2 | 2 |
| Sampled future frames | 6 | 7 | 6 | 6 | 6 | 6 |
| Learning rate | $5 \times 10^{-4}$ | $1 \times 10^{-4}$ | $1 \times 10^{-4}$ | $1 \times 10^{-4}$ | $5 \times 10^{-4}$ | $1 \times 10^{-4}$ |
| LR Schedule | | | Constant | | | |
| Weight decay | | | 0.0 | | | |
| Grad clip | | | 1.0 | | | |
| Warmup steps | | | 500 | | | |
| Loss balancing | | | Equal weights | | | |
| Optimizer | | | AdamW | | | |
| Mixed precision | | | bf16 | | | |

| Transformer | Pre-train | BAIR | RoboNet | VP$^2$ | Pre-train | RoboNet |
|---|---|---|---|---|---|---|
| GPU days | 19 | 1.5 | 10 | 3 | 9 | 26 |
| Training steps | $7 \times 10^5$ | $1 \times 10^5$ | $6 \times 10^5$ | $2 \times 10^5$ | $3.5 \times 10^5$ | $5 \times 10^5$ |
| Batch size | 64 | 64 | 64 | 64 | 16 | 32 |
| Sequence length | 16 | 16 | 12 | 12 | 16 | 12 |
| Context frames | 2 | 1 | 2 | 2 | 2 | 2 |
| Learning rate | | | $1 \times 10^{-4}$ | | | |
| LR Schedule | | | Cosine | | | |
| Weight decay | | | 0.01 | | | |
| Grad clip | | | 1.0 | | | |
| Warmup steps | | | 5000 | | | |
| Loss balancing | | | N/A or equal weights | | | |
| Optimizer | | | AdamW | | | |
| Mixed precision | | | bf16 | | | |
| Sampling temperature | | | 1.0 | | | |
| Sampling top-$k$ | | | 100 | | | |

block, $F_p^l$ is augmented with $F_c^l$ as follows:

$$
\begin{aligned}
F_c^{l+1} &= \text{EncBlock}_c^{l+1}(F_c^l) \\
F_p^{l+1} &= \text{EncBlock}_p^{l+1}(\text{Augment}(F_p^l, F_c^l))
\end{aligned}
\tag{4}
$$

This is achieved by performing cross-attention between the $2hw$ positions of the feature maps:

$$
\begin{aligned}
\text{Flatten: } Q &= \text{Norm}\left(\text{Reshape}\left(F_p^l\right)\right) + \text{PosEmb}^Q \in \mathbb{R}^{hw \times c} \\
K = V &= \text{Norm}\left(\text{Reshape}\left(F_c^l\right)\right) + \text{PosEmb}^{KV} \in \mathbb{R}^{hw \times c} \\
\text{Cross-Attention: } R &= \text{Attention}\left(QW^Q, KW^K, VW^V\right) \in \mathbb{R}^{hw \times c} \\
\text{Residual-Connection: } \text{Augment}(F_p^l, F_c^l) &= \text{SiLU}\left(F_p^l + \text{Reshape}(R)\right) \in \mathbb{R}^{c \times h \times w}.
\end{aligned}
\tag{5}
$$

To reduce memory usage, we apply the cross-attention mechanism only when the feature map size is below a certain threshold ($16 \times 16$ for a $64 \times 64$ original resolution and $32 \times 32$ for a $256 \times 256$ resolution). This mechanism is symmetrically performed across the context and prediction decoder.

Since attention mechanisms can flexibly handle varied input lengths, the conditioning mechanism can be easily extended to accommodate different numbers of context frames. Each context frame is

independently processed by the context encoder and decoder, and their feature maps are concatenated to serve as inputs for cross-attention in the prediction encoder and decoder.

Our VQGAN for $256 \times 256$ resolution is initialized from the pretrained model from aMUSEd[4] [73]. We do not use discriminators for $64 \times 64$ resolution, effectively converting the VQGAN into a vanilla VQVAE with an additional perceptual loss.

**Transformer.** We flatten a video into a sequence of tokens:

$$x = (z_1^{(1)}, \ldots, z_1^{(N)}, [\text{S1}], z_2^{(1)}, \ldots, z_2^{(N)}, \ldots, [\text{S2}], z_{T_0+1}^{(1)}, \ldots, z_{T_0+1}^{(n)}, \ldots), \quad (6)$$

where we use two types of slot tokens [S1] and [S2] before the start of context frames and future frames, respectively. Context and future frames do not share token IDs, resulting in a transformer vocabulary of 16,386 tokens: the first 8,192 for context frames, the next 8,192 for future frames, and the last two for slot tokens. We adopt the autoregressive transformer architecture from LLaMA [87], but instantiate it to smaller models matching the size of GPT-2. We considered two model sizes, listed in Figure 2. Most of our experiments utilize a 138M parameter transformer, while preliminary scaling analysis is conducted using a 436M model.

## A.2 Action-free Video Pre-training

**Data mixture.** We pre-train iVideoGPT using 35 datasets from the Open X-Embodiment Dataset (OXE) [70] and Something-Something-v2 (SSv2) [25]. To construct our training dataset from OXE, we implement a filtering and weighting process similar to Octo [67]. Initially, we exclude datasets lacking image streams and those derived from mobile robots. Subsequently, datasets exhibiting excessive repetition or possessing low image resolutions are eliminated. The remaining datasets were categorized as either "large" or "small," and each was assigned a weight based on its size and diversity. We select $1\%$ of samples from each subset as validation data and use the rest for training. For SSv2, we manually select 95 classes with clear motion trends from the original 174 video classes as our pre-training data with a weighting of 15%. We use the official splits of SSv2 for training and validation. For a comprehensive breakdown of the mixture, refer to Table 4.

**Training details.** During training, we sample sequences of frames by first randomly selecting a training video and then uniformly sampling a segment of a specified length and step size, i.e., neighboring frames in the segment are spaced a certain number of steps apart in the original video. We observe that datasets are collected at different frequencies. To maintain consistency, we adjust sampling with varied step sizes, aligning each with its respective dataset frequency, as listed in Table 4. For tokenizer training, the initial frames of the segment are used as context frames, and from the remaining frames, we randomly sample a subset as future frames to reduce memory requirements and increase batch size. For transformer training, we use the full segment of frames. The number of frames in minibatches for each dataset is detailed in Table 3. We use a mixture of OXE and SSv2 for training the tokenizer to ensure visual diversity, while only OXE is used for training the transformer. For data augmentation, we apply random resized crop and color jitter, ensuring consistency across the sequence. During both tokenizer and transformer training, we blend different losses with equal weights. Unless specified otherwise, we follow the same implementation details when fine-tuning iVideoGPT on downstream tasks.

**Goal-conditioned prediction.** To train a goal-conditioned variant of iVideoGPT on the same dataset, we first fine-tune the previously obtained tokenizer using two randomly sampled frames as context for 550k training steps. Then, we train a transformer from scratch with the rearranged frame segment $\tilde{o}_{1:T} = (o_T, o_1, o_2, \ldots, o_{T-1})$ for 1 million steps. The architecture and training procedures remain consistent with the above setup.

**License.** The Open X-Embodiment dataset follows the Apache license. RoboNet is licensed under Creative Commons Attribution 4.0, while BAIR follows Creative Commons BY 4.0. The Something-Something-V2 dataset is subject to the Data License Agreement.

---

[4]`https://github.com/huggingface/amused` under openrail++ license

Table 4: iVideoGPT pre-training data mixture from the Open X-Embodiment [70] and Something-Something-V2 [25] datasets.

| Dataset | Num of trajectories | Step size | Sampling weight |
|---|---|---|---|
| Fractal (RT-1) [10] | 87,212 | 1 | 12.8% |
| Bridge [94] | 28,935 | 2 | 12.8% |
| BC-Z [41] | 43,264 | 3 | 12.8% |
| RoboNet [16] | 82,649 | 1 | 12.8% |
| Kuka [46] | 580,392 | 3 | 8.5% |
| Language Table [56] | 442,226 | 3 | 4.2% |
| Stanford MaskViT [27] | 9,200 | 1 | 4.2% |
| UIUC D3Field [95] | 768 | 1 | 2.2% |
| Taco Play [78, 61] | 3,603 | 5 | 0.5% |
| Jaco Play [17] | 1,085 | 3 | 0.5% |
| Roboturk [58] | 1,995 | 3 | 0.5% |
| Viola [115] | 150 | 7 | 0.5% |
| Toto [111] | 1,003 | 10 | 0.5% |
| Columbia Cairlab Pusht Real [14] | 136 | 3 | 0.5% |
| Stanford Kuka Multimodal Dataset [54] | 3,000 | 7 | 0.5% |
| Stanford Hydra Dataset [6] | 570 | 3 | 0.5% |
| Austin Buds Dataset [116] | 50 | 7 | 0.5% |
| NYU Franka Play Dataset [15] | 456 | 1 | 0.5% |
| Furniture Bench Dataset [35] | 5,100 | 3 | 0.5% |
| UCSD Kitchen Dataset [100] | 150 | 1 | 0.5% |
| UCSD Pick and Place Dataset [23] | 1,355 | 1 | 0.5% |
| Austin Sailor Dataset [65] | 240 | 7 | 0.5% |
| UTokyo PR2 Tabletop Manipulation [68] | 240 | 3 | 0.5% |
| UTokyo Xarm Pick and Place [59] | 102 | 3 | 0.5% |
| UTokyo Xarm Bimanual [59] | 70 | 3 | 0.5% |
| KAIST Nonprehensile [49] | 201 | 3 | 0.5% |
| DLR SARA Pour [72] | 100 | 3 | 0.5% |
| DLR SARA Grid [71] | 107 | 3 | 0.5% |
| DLR EDAN Shared Control [92, 74] | 104 | 3 | 0.5% |
| ASU Table Top [113, 112] | 110 | 4 | 0.5% |
| UTAustin Mutex [82] | 1,500 | 7 | 0.5% |
| Berkeley Fanuc Manipulation [114] | 415 | 3 | 0.5% |
| CMU Playing with Food [79] | 174 | 3 | 0.5% |
| CMU Play Fusion [13] | 576 | 2 | 0.5% |
| CMU Stretch [5, 62] | 135 | 3 | 0.5% |
| Something-Something-V2 [25] | 120,581 | 1 | 15.0% |
| Total | 1,417,954 | - | 100.0% |

---

**Algorithm 1** Model-Based Policy Optimization (MBPO), adapted from [42]

---

1: Initialize actor-critic $\pi_\phi, v_\psi$, world model $p_\theta$
2: Initialize real replay buffer $\mathcal{D}_{\text{real}}$ with random policy
3: Initially train model $p_\theta$ on $\mathcal{D}_{\text{real}}$
4: Initialize imagined replay buffer $\mathcal{D}_{\text{imag}}$ with random rollouts using $p_\theta$
5: **for** $N$ steps **do**
6:     // Training
7:     **if** model update step **then**
8:         Update world model $p_\theta$ on a mini-batch from $\mathcal{D}_{\text{real}}$
9:     **end if**
10:     Update actor-critic $\pi_\phi, v_\psi$ with model-free objectives on a mini-batch from $\mathcal{D}_{\text{imag}} \cup \mathcal{D}_{\text{real}}$
11:     // Data collection
12:     **if** model rollout step **then**
13:         Sample a mini-batch of $o_t$ uniformly from $\mathcal{D}_{\text{real}}$
14:         Perform $k$-step model rollout starting from $o_t$ using policy $\pi_\phi$; add to $\mathcal{D}_{\text{imag}}$
15:     **end if**
16:     Take action in environment according to $\pi_\phi$; add to $\mathcal{D}_{\text{real}}$
17: **end for**

---

### A.3 Video Prediction

**Evaluation metrics.** We evaluate our model across four different metrics[5]: Structural Similarity Index Measure (SSIM) [97], Peak Signal-to-noise Ratio (PSNR) [40], Learned Perceptual Image Patch Similarity (LPIPS) [108] and Fréchet Video Distance (FVD) [88]. Following prior works [3, 91, 4, 104], we account for the stochastic nature of video prediction by sampling 100 future trajectories per test video and selecting the best one for the final PSNR, SSIM, and LPIPS scores. For FVD, we use all 100 samples.

### A.4 Visual Planning

We use the official repository[6] to evaluate our model on the $\text{VP}^2$ benchmark. The reported baseline results are provided by the authors of the benchmark. For the Robosuite tasks, a cost below $0.05$ is considered a success.

### A.5 Visual Model-based RL

**Environments.** Meta-world [105], following MIT License, is a benchmark of 50 robotic manipulation tasks. We select six tasks for our experiments: Button Press Topdown Wall, Plate Slide, Hammer, Door Lock, Handle Pull Side, and Coffee Push. We set the maximum episode length to 200 environment steps with an action repeat of 2 and a frame stack of 3 across these tasks and adjust the number of training steps to match the varying difficulty levels. During experiments, we observed that high rewards do not consistently correlate with high success rates in the original Meta-world implementation. This discrepancy presents a challenge to the learning stability of agents. To address this, we introduce an additional bonus for task success $r_{\text{bonus}} = 10.0$ alongside the original task reward $r_{\text{task}}$:

$$r = r_{\text{task}} + r_{\text{bonus}} \cdot \mathbb{I}_{\text{task success}}. \tag{7}$$

Moreover, Meta-world features hard-exploration tasks, resulting in significant variance in the learning curves, which poses challenges to the accurate evaluation of RL algorithm performance. To mitigate this issue, we initialize the replay buffer of all compared methods, with 5 successful demonstration trajectories for all tasks except Door Lock. This strategy, commonly used to accelerate reinforcement learning [36], helps stabilize the training process and provides a more reliable evaluation.

---

[5]We use public implementations of metrics: `https://github.com/francois-rozet/piqa` under MIT license for SSIM and PSNR, `https://github.com/richzhang/PerceptualSimilarity` under BSD-2-Clause license for LPIPS, and `https://github.com/universome/stylegan-v` under NVidia license for FVD.

[6]`https://github.com/s-tian/vp2`

Table 5: Hyperparameters of model-based RL with iVideoGPT.

| Model-based RL | Hyperparameter | Value |
|---|---|---|
| Model rollout | Init rollout batch size | 640 |
| | Interval | 200 env. steps |
| | Batch size | 32 |
| | Horizon | 10 |
| Model training | Init training steps | 1000 |
| | Tokenizer training interval | 40 env. steps |
| | Transformer training interval | 10 env. steps |
| | Batch size | 16 |
| | Sequence length | 12 |
| | Context frames | 2 |
| | Sampled future frames (tokenizer) | 5 |
| | Learning rate | $1 \times 10^{-4}$ |
| | Weight decay | 0 |
| | Optimizer | Adam |
| Model-based RL | Real data ratio | 0.5 |

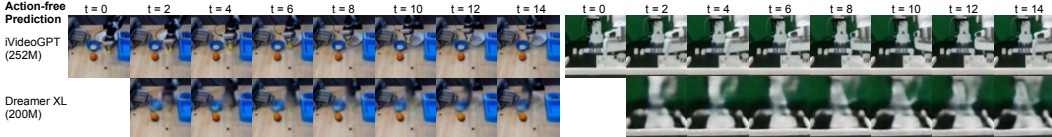

Figure 11: Qualitative evaluation on the Open X-Embodiment dataset for Dreamerv3-XL pre-trained on the same dataset as ours.

**Implementation Details.** We have developed a simple model-based RL algorithm using iVideoGPT as a world model within the MBPO [42] framework, with DrQ-v2 [103] as the base actor-critic RL algorithm. Please refer to Algorithm 1 for the pseudo-code. Our implementation is based on the official DrQ-v2 code[7], using the same hyperparameters and architecture for actor-critic learning. Hyperparameters specific to model-based RL are listed in Table 5. We use a symlog transformation [32] for reward prediction in iVideoGPT.

**Baselines.** To compare our method with DreamerV3, which lacks native pre-training support, we use APV [81]—a method enabling action-free pre-training on DreamerV2—as a baseline, modified to incorporate DreamerV3 features. We pre-train this model on the same dataset as iVideoGPT.

## B  Extended Experimental Results

### B.1  Qualitative Evaluation

We present additional examples of video predictions by iVideoGPT on various datasets in Figures 12, 13, 15, 16, 17, 18, and 19. We also include an additional showcase of zero-shot predictions by the pre-trained transformer in iVideoGPT in Figure 14, supplementing Figure 8 of the main text.

Additionally, we showcase prediction examples from the large-scale pre-trained DreamerV3-XL on the Open X-Embodiment dataset in Figures 11.

### B.2  Human Study

Numerical metrics like FVD don't always align with human-judged visual quality. To address this, we conduct a human user study on the prediction results of various models. Due to the lack of official

---

[7]`https://github.com/facebookresearch/drqv2` under MIT License

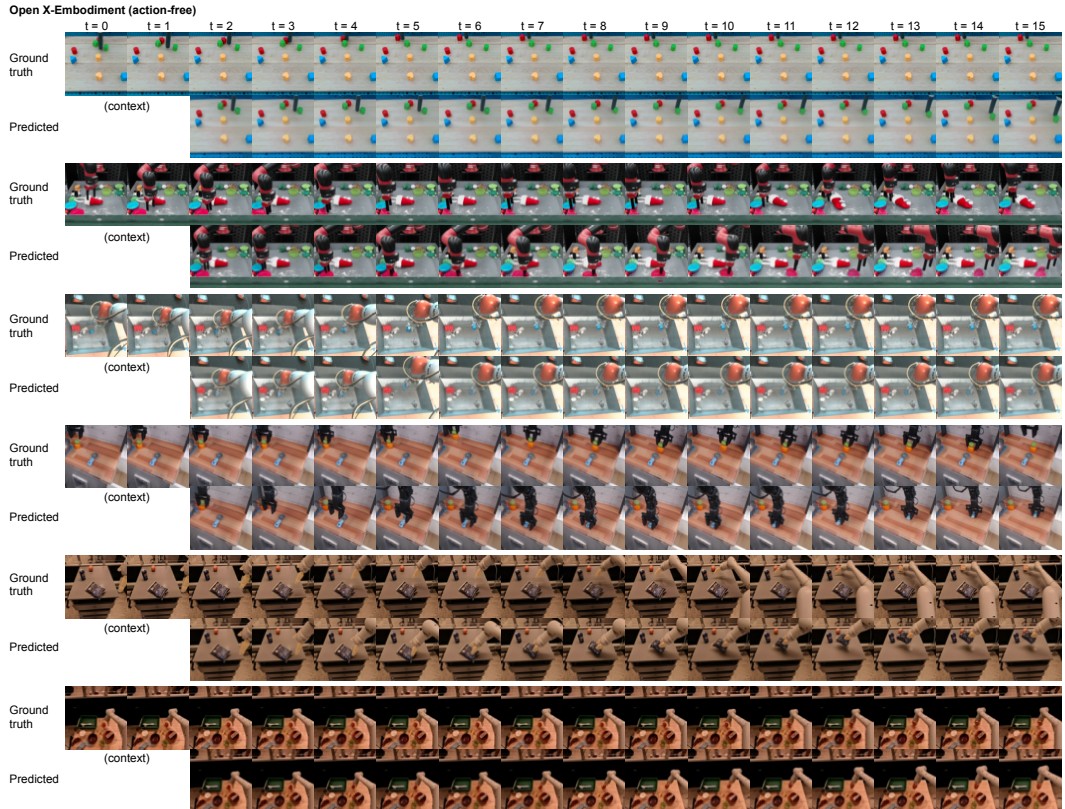

Figure 12: Additional qualitative evaluation on the Open X-Embodiment dataset for action-free video prediction.

pretrained models for most baselines, we are only able to compare iVideoGPT with VideoGPT [101] and MCVD [93] on the action-free BAIR dataset. We generate videos using these three models from the test set and ask users to label preferences between two randomly sampled videos based on the physical naturalness and feasibility of robot-object interactions. A total of 386 annotations are collected from 9 participants. The results in Figure 20 demonstrate that iVideoGPT is preferred by human annotators more.

### B.3 Visual Planning

Quantitative results on the VP$^2$ benchmark are reported in Table 6.

### B.4 Visual Model-based RL

**Comparison to FitVid-based world models.** Although FitVid [4] is originally designed for the video prediction task and has not been used as world models for MBRL, we have implemented a baseline using FitVid by replacing iVideoGPT in our implementation. To predict rewards, we add an MLP head on top of FitVid's latent states, parallel to the image decoder. As shown in Figure 21, MBPO with iVideoGPT outperforms FitVid on 5 out of 6 tasks and performs comparably on the remaining one. We also qualitatively observe that FitVid's imagined trajectories are blurrier compared to ours, which hinders its ability to simulate real environments accurately and may hurt MBRL performance.

### B.5 Computational Efficiency

We report training and inference time and memory usage with various tokenizers in Table 7 and 8, respectively. Our proposed compressive tokenization provides significant memory savings during training and faster rollouts during generation. We note that although we use a more complicated

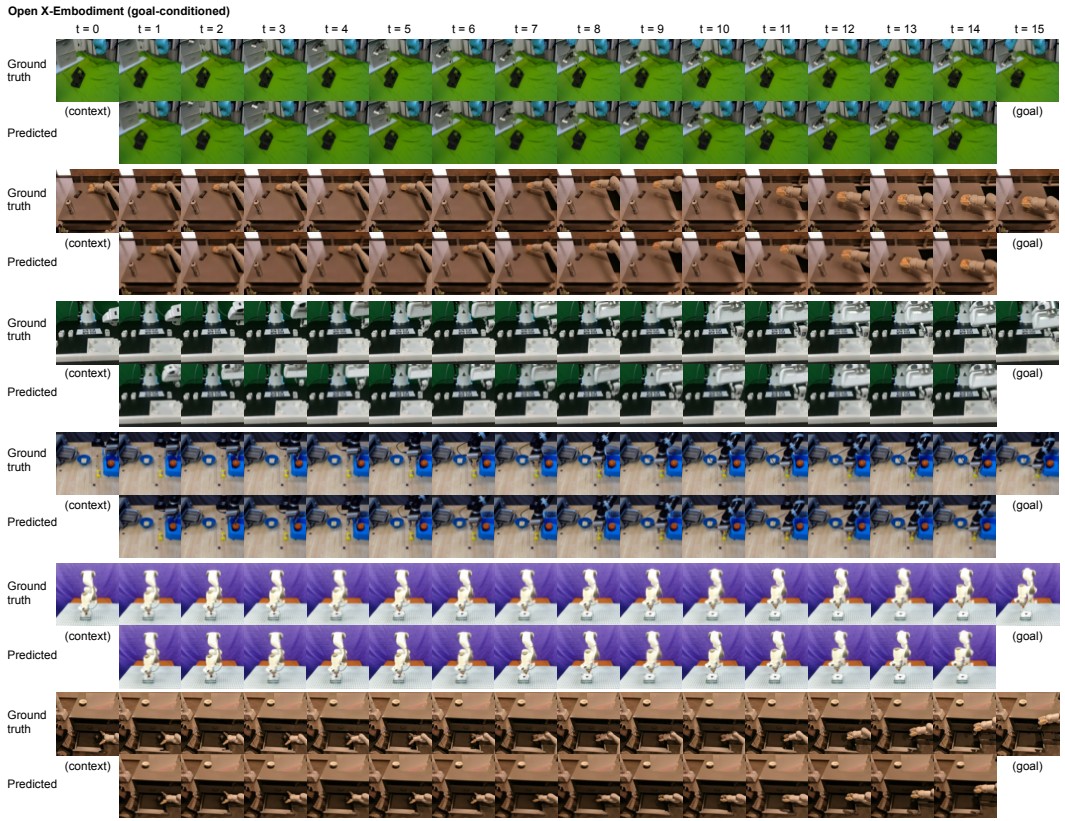

Figure 13: Additional qualitative evaluation on the Open X-Embodiment dataset for goal-conditioned video prediction.

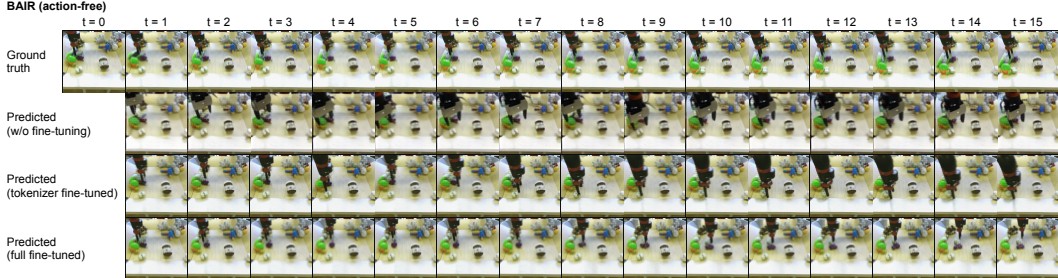

Figure 14: Additional zero-shot prediction by the pre-trained transformer in iVideoGPT, supplementing Figure 8 of the main text.

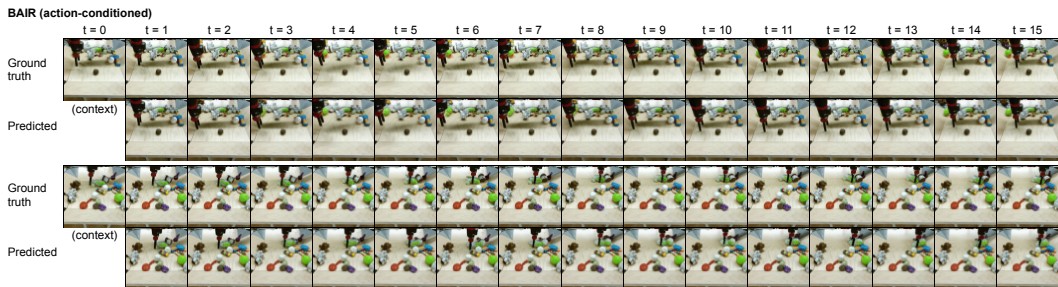

Figure 15: Additional qualitative evaluation on the BAIR dataset, given future actions.

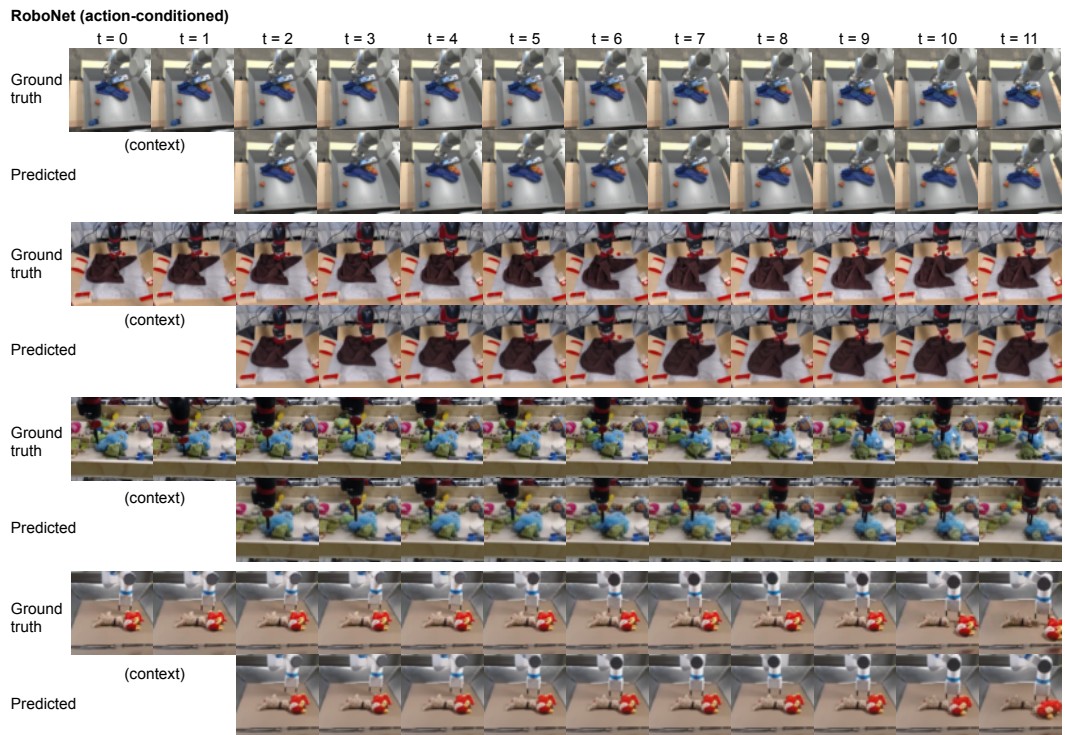

Figure 16: Additional qualitative evaluation on the RoboNet dataset, highlighting accurate movements of the pushed objects.

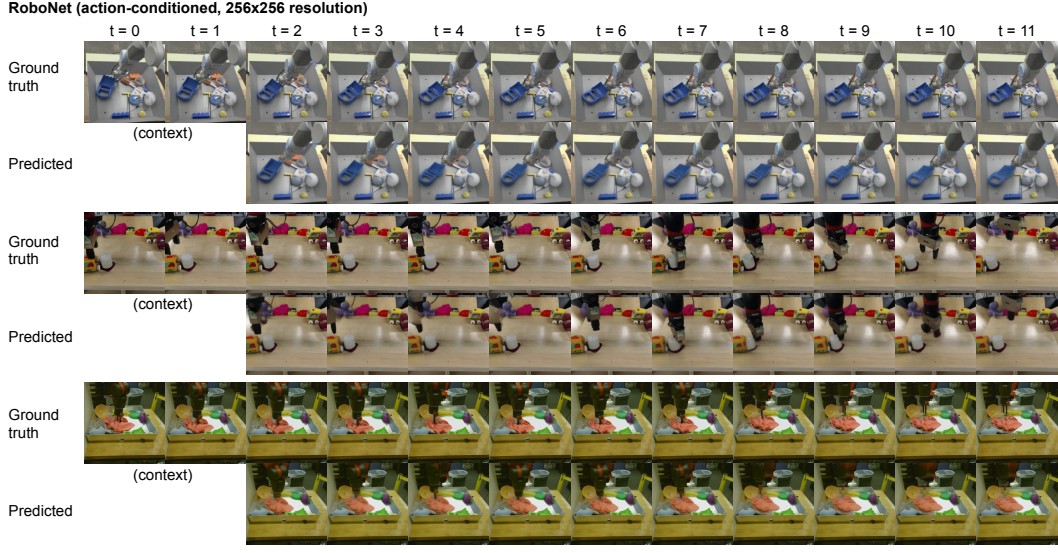

Figure 17: Additional qualitative evaluation on the RoboNet dataset, in high resolution ($256 \times 256$).

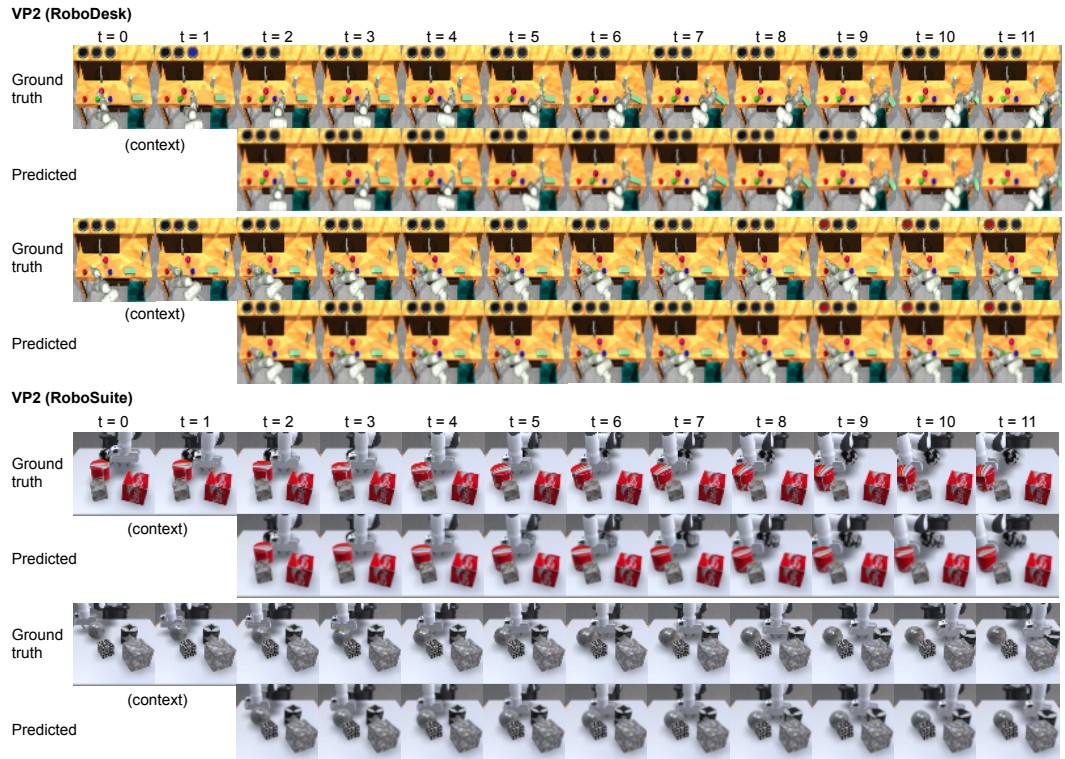

Figure 18: Additional qualitative evaluation on the VP$^2$ benchmark.

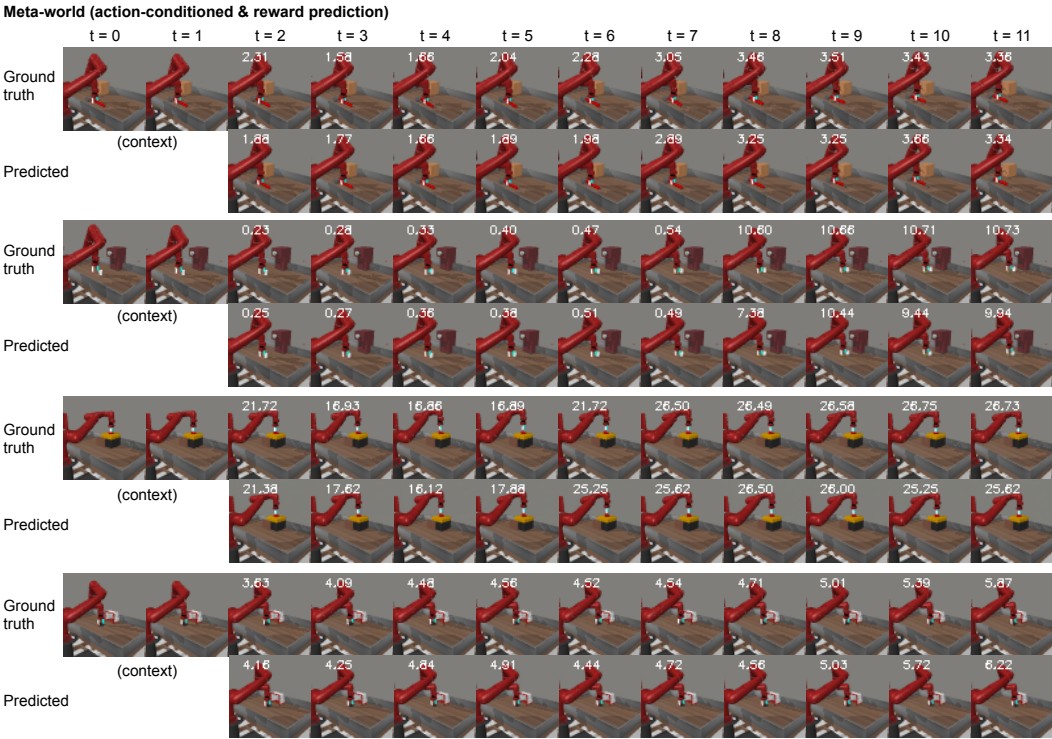

Figure 19: Additional qualitative evaluation on Meta-world tasks. True and predicted rewards are labeled at the top left corner. Zoom in for details.

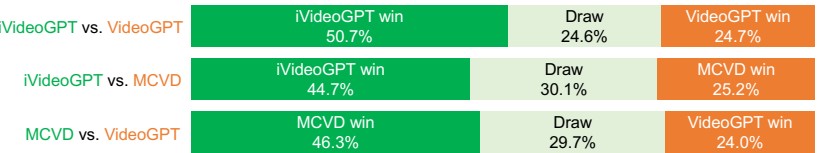

Figure 20: Human study. Videos generated by three models, VideoGPT, MCVD, and iVideoGPT, on the action-free BAIR dataset are presented to human users, who label their preferences based on the physical naturalness and feasibility of robot-object interactions.

Table 6: Quantitative results on the VP$^2$ benchmark, reporting mean, min, and max performance over various control runs.

| Tasks | Success rate | iVideoGPT (ours) | FitVid | SVG′ | MCVD | MaskViT | Struct-VRNN | Simulator |
|---|---|---|---|---|---|---|---|---|
| Robosuite push | mean | 0.7833 | 0.6760 | 0.7980 | 0.7733 | **0.8260** | 0.5540 | 0.9350 |
| | max | 0.7950 | 0.7900 | 0.8400 | 0.7900 | 0.8500 | 0.6000 | 0.9500 |
| | min | 0.7750 | 0.6400 | 0.7600 | 0.7400 | 0.7900 | 0.5000 | 0.9200 |
| Flat block | mean | 0.0333 | **0.0917** | 0.0200 | 0.0500 | 0.0400 | 0.0467 | 0.1333 |
| | max | 0.0417 | 0.1333 | 0.0333 | 0.0667 | 0.1000 | 0.1333 | 0.1333 |
| | min | 0.0250 | 0.0667 | 0.0000 | 0.0333 | 0.0000 | 0.0000 | 0.1333 |
| Open drawer | mean | **0.3750** | 0.2533 | 0.1667 | 0.1167 | 0.0400 | 0.0267 | 0.7667 |
| | max | 0.3917 | 0.3333 | 0.2667 | 0.1333 | 0.1000 | 0.1000 | 0.7667 |
| | min | 0.3500 | 0.1333 | 0.0667 | 0.1000 | 0.0000 | 0.0000 | 0.7667 |
| Open slide | mean | 0.1611 | 0.3533 | **0.5733** | 0.1833 | 0.0867 | 0.1267 | 0.7167 |
| | max | 0.1917 | 0.4000 | 0.7333 | 0.2000 | 0.1667 | 0.2333 | 0.7333 |
| | min | 0.1250 | 0.2667 | 0.4667 | 0.1667 | 0.0333 | 0.0667 | 0.7000 |
| Blue button | mean | 0.9556 | 0.9400 | **0.9733** | 0.9500 | 0.9467 | 0.8667 | 1.0000 |
| | max | 0.9833 | 0.9667 | 1.0000 | 1.0000 | 0.9667 | 0.9000 | 1.0000 |
| | min | 0.9333 | 0.8667 | 0.9333 | 0.9000 | 0.9333 | 0.8000 | 1.0000 |
| Green button | mean | 0.8250 | **0.8400** | 0.8133 | 0.8333 | 0.6400 | 0.6800 | 0.9667 |
| | max | 0.8667 | 0.9000 | 0.9000 | 0.8333 | 0.7000 | 0.8000 | 0.9667 |
| | min | 0.7833 | 0.7667 | 0.7667 | 0.8333 | 0.6000 | 0.5667 | 0.9667 |
| Red button | mean | **0.9222** | 0.5867 | 0.7600 | 0.7333 | 0.2400 | 0.3067 | 0.9000 |
| | max | 0.9333 | 0.6333 | 0.8667 | 0.7333 | 0.3333 | 0.3333 | 0.9000 |
| | min | 0.9000 | 0.5000 | 0.6333 | 0.7333 | 0.1333 | 0.2333 | 0.9000 |
| Upright block | mean | 0.4472 | 0.5133 | 0.4867 | 0.5667 | **0.6200** | 0.3333 | 0.9000 |
| | max | 0.4667 | 0.5667 | 0.6667 | 0.6000 | 0.7333 | 0.3667 | 0.9000 |
| | min | 0.4250 | 0.5000 | 0.4000 | 0.5333 | 0.5000 | 0.3000 | 0.9000 |

tokenizer design, it is not the bottleneck of generation time. Additionally, although we use more tokens for context frames compared to the $4 \times 4$ tokenizer, generation time is primarily influenced by the number of forward passes of the autoregressive transformer, which remains the same.

Table 7: Training efficiency of the transformer with various tokenizers, measured on 40G A100 GPUs with a per-device batch size of 16.

| Tokenizer | Speed (#iters/sec) | Memory (GB) |
|---|---|---|
| $4 \times 4$ | 3.10 | 10.6 |
| $16 \times 16$ | N/A | **OOM** |
| Ours | 2.62 | 22.3 |

Table 8: Generation efficiency with various tokenizers, measured on an RTX 4090 GPU with a batch size of 1.

| Tokenizer | Tokenize (sec) | Generation (sec) | Detokenize (sec) | Memory (GB) |
|---|---|---|---|---|
| $4 \times 4$ | 0.27 | 1.13 | 0.05 | 1.98 |
| $16 \times 16$ | 0.26 | **22.5** | 0.04 | 1.90 |
| Ours | 0.29 | 1.11 | 0.06 | 2.33 |

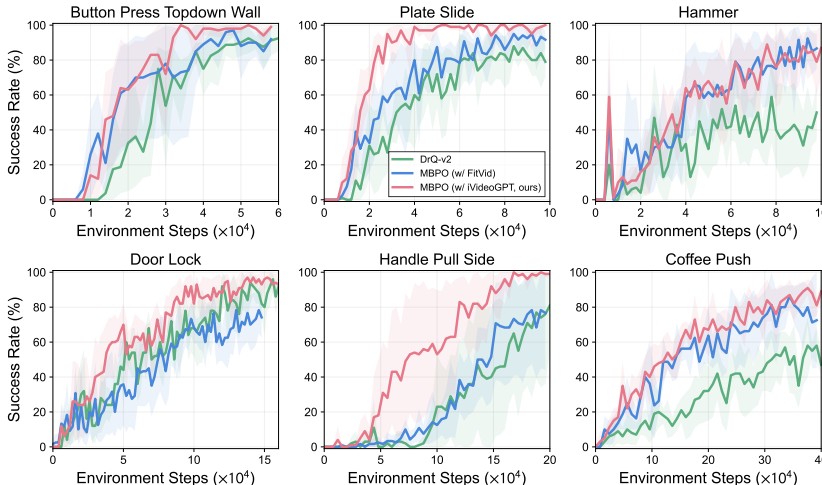

Figure 21: Visual model-based RL on Meta-world, comparing to an additional baseline using FitVid as world models.

## C   Extended Discussions

### C.1   Differences with IRIS

Discrete tokenization and autoregressive transformers are prevalent in contemporary deep learning due to their simplicity and generality. iVideoGPT generally shares this architecture with IRIS [63], but possesses distinguishing features, summarized as follows:

- **Pre-training and fine-tuning paradigm**: iVideoGPT is designed for a paradigm that involves pre-training on large-scale videos and fine-tuning on various downstream tasks. In contrast, IRIS focuses solely on MBRL with Transformer-based world models trained from scratch in the Atari domain.

- **Efficient tokenization**: iVideoGPT proposes novel compressive tokenization to significantly reduce the number of tokens, saving time and memory (see Table 7 and 8), while IRIS uses per-frame tokenization.

- **Flexible action-conditioning design**: iVideoGPT employs slot tokens with optional additive action embeddings to support both action-free pre-training and action-conditioned fine-tuning, while IRIS strictly treats discrete Atari actions as tokens.

- **Off-policy MBRL implementation**: iVideoGPT uses an off-policy RL algorithm while IRIS performs on-policy learning. On-policy learning needs a large number of model rollouts, which, combined with inefficient tokenization, results in 7 days for 100k-environment-step training. In comparison, iVideoGPT only needs ~4 hours.

### C.2   Differences with VideoGPT

We elaborate on the the difference between the tokenizer in VideoGPT [101] and ours, and how they impact interactivity.

VideoGPT uses a VQVAE for video that relies on a series of 3D convolutions to downsample across space and time. For example, it downsamples original pixels from $16 \times 64 \times 64$ to discrete tokens of $8 \times 32 \times 32$ or $4 \times 16 \times 16$, depending on the downsampling ratio. The key issue is that this non-causal downsampling over the temporal dimension results in each token containing information from a window of frames. As a result, the entire video of a fixed length can only be reconstructed after VideoGPT generates all tokens. As shown in Figure 2, VideoGPT only allows the input of future action sequences at the beginning of prediction, preventing an agent from interactively determining its actions based on predicted observations. In contrast, our tokenizer discretizes video frames separately, using a conditional mechanism to handle temporal redundancy, enabling frame-by-frame video generation and allowing for intermediate action intervention.

Moreover, our tokenizer's novel design, with its cross-attention mechanism, is more efficient in handling temporal redundancy, converting videos into significantly fewer tokens ($L = 511$ with $N = 256, n = 16, T = 16, T_0 = 1$ as stated in Section 3.1). In contrast, VideoGPT finds that using a larger downsampling ratio than a token size $8 \times 32 \times 32$, results in worse performance.

### C.3  Failure Case Analysis for Visual Planning

Our model performs sub-optimally on the RoboDesk open slide task from the VP$^2$ benchmark. In this section, we investigate the underlying causes through case studies, attributing the performance issues to limitations in both our model and the benchmark.

**Inaccurate model prediction.**  Despite achieving excellent overall video prediction metrics, such as mean square error and perceptual loss, on the validation set for the open slide task, our model predicts wrong outcomes on a few trajectories. We visualize these trajectories in Figure 22 and find that while the observation is limited to $64 \times 64$ resolution, the task of opening the slide requires the model to capture subtle changes, particularly whether the robot's gripper has made contact with the slide handle. Actually, even humans struggle to discriminate this detail with low-resolution inputs. Due to this uncertainty, the model may incorrectly predict a sequence of imprecise actions as successful. This overconfidence [66] can be exacerbated in the process of model predictive control, which samples a large number of action candidates and selects the "best" one according to the model. Our analysis provides an explanation to the observation by Tian et al. [86] that overall excellent perceptual metrics do not always correlate with effective control performance, as the worst-case scenarios are critical in model-predictive control.

Furthermore, we hypothesize that our two-stage architecture of tokenization and prediction can exacerbate the aforementioned uncertainty, as discrete tokenization inevitably results in some loss of information from the observations. This hypothesis is supported by the fact that end-to-end models like SVG′ [91] and FitVid [4] perform significantly better than two-stage models, including ours and others like MaskViT [27], which uses a visual tokenizer, and Struct-VRNN [64], which employs a keypoint-based representation.

We anticipate that training and evaluating our model at a higher resolution, such as $256 \times 256$, could mitigate these issues and enhance control performance. However, we currently conduct experiments at a lower resolution to ensure a fair comparison with other models.

**Imperfect built-in reward design.**  We observe that no current model in the VP$^2$ benchmark consistently outperforms other models across all tasks, and iVideoGPT is no exception. Beyond models' inaccuracies in prediction for severely out-of-distribution (OOD) actions, our analysis of this inconsistent performance also reveals flaws in the benchmark's built-in reward design.

In VP$^2$, scores for sampled actions are estimated mainly by a learned classifier that assesses task success based on model-predicted frames. This classifier, trained by the VP$^2$ authors, appears to lack robustness and is easily fooled by OOD inputs, assigning high rewards to low-quality or unlikely-to-succeed predicted trajectories (see examples in Figure 23). Such an imperfect reward function likely contributes to the mixed results observed on this benchmark, with iVideoGPT even outperforming the oracle simulator in one task. Addressing visual planning with imperfect rewards is an independent research problem and beyond the scope of this paper.

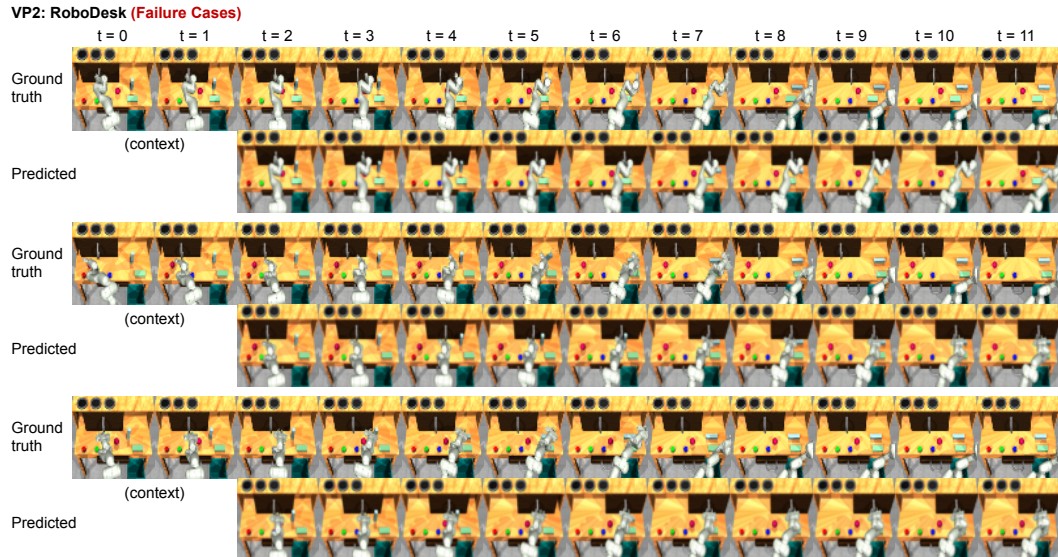

Figure 22: Failure case analysis on the RoboDesk open slide task from the VP$^2$ benchmark, where, likely due to the low resolution of observations, our model fails to discriminate between subtle changes, particularly whether the robot's gripper has made contact with the slide handle.

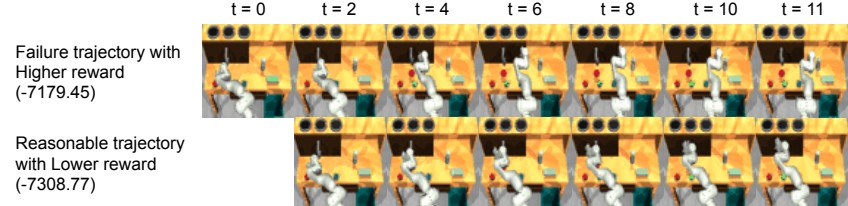

Figure 23: Imperfect built-in reward in VP$^2$ benchmark. A learned reward model can assign high rewards to predicted transitions that are less likely to succeed, which can mislead optimizers in model-predictive control.

## D   Computational Resources

We implement iVideoGPT in PyTorch, using the `diffusers`[8] and `transformers`[9] libraries. Our models are trained and evaluated on an A100 and RTX 3090 GPU cluster. Each experiment utilizes 4 GPUs in parallel, with 16 data loader workers per device. GPU days required for training are reported in Table 2. Experiments at $64 \times 64$ resolution can be conducted with 24 GB of GPU memory per device, while $256 \times 256$ resolution requires 40 GB. The Open X-Embodiment dataset is particularly large, occupying about 5TB of disk space.

## E   Broader Impact

World models advance the development of autonomous machine intelligence, particularly through the valuable visual insights offered by videos. However, their full potential remains untapped without scalable and interactive architectures capable of distilling vast amounts of commonsense knowledge from multimodal data. This paper, we believe, takes an important step by introducing a flexible framework with a specific focus on the robotic manipulation domain. Our results may pave the way for higher-quality world models applicable across diverse domains, enhancing performance in control tasks of embodied intelligence. Despite the benefits, designing and training these models is challenging, requiring substantial computational power and increasing the carbon footprint. Using

---

[8]`https://github.com/huggingface/diffusers` under Apache License

[9]`https://github.com/huggingface/transformers` under Apache License

underdeveloped, inaccurate world models for autonomous control could lead to risky actions and potential physical damage, which can be mitigated by developing uncertainty-aware models to prevent uncertain actions. Conversely, the advancement of these models could lead to job displacement in sectors relying on manual control tasks. Additionally, the underlying techniques for world models can be misused to generate synthetic videos that mimic real events or people. However, since our model is merely a research prototype, trained only with robotic and human manipulation data and relatively small in scale, we do not anticipate immediate negative societal impacts such as deepfakes or job displacement.

