# OpenReview forum: "iVideoGPT: Interactive VideoGPTs are Scalable World Models"
_NeurIPS.cc/2024/Conference — NeurIPS 2024 poster_

### Official Review · Reviewer_6ajb · 2024-07-11

**Soundness:** 2
**Presentation:** 2
**Contribution:** 4
**Rating:** 6
**Confidence:** 3

**Summary:**

This work studies the setting of planning and prediction from video world models. It proposes to use a GPT-style transformer world model that incorporates action and reward information in it's context and prediction pipeline. The model is further equipped with a novel tokenization technique based on VQGAN. Finally, the model is evaluated on three downstream applications, namely video prediction, planning and model-based RL. Ablations on the models and tokenizers scalability are conducted.

**Strengths:**

Problem Statement
* The problem of studying dyna-style video algorithms with foundational models is interesting and warrants investigation.

Clarity
* The paper is very well written and the language is clear
* The paper follows a very concrete thread and is easy to follow
* All plots and figures are well-formatted and easy to understand

Related Work
* The treatment of related work is done very nicely. The paper cites a total of 120 other works which is not often seen and thus formidable.
* The work highlights older as well as recent work and positions itself well within the literature

Experiments
* The experiments contain a large number of baselines and comparisons that are useful to understand the capabilities of the model for downstream return maximization.

**Weaknesses:**

Clarity
* In some cases, the writing makes very strong statements that are very broad and hard to be supported by evidence. E.g.
  * L97 "This model can acquire broad world knowledge". This statement is not supported and it is unclear to me what "broad world knowledge" means. I believe concise language about the capabilities of our algorithms is important.

Motivation
* One issue with the paper is that some of the motivation is not quite clear to me. The paper argues that the provided world model has two benefits: It is interactive and scalable.
  * First, it is not clear to me what it means for this model to be interactive. If the condition for a model is to be action-conditioned, then the dreamer world-model should also be interactive. In fact, even much older world-models would be interactive [1, 2].
  * Second, it is not clear to me that other models would not be scalable and the paper provides no evidence that they are not scalable. The text states in line 91 that dreamer lacks scability, however not citation or evidence for this is provided. For instance, dreamer uses state-space models (SSMs) and recent advances have built scalable SSMs [1].

Experiments and Claims
* For the tokenizer, the text claims that it "decouples dynamics information from context conditions" L161. This seems like a strong statement that would warrant evidence to be supported. I believe qualitatively looking at a small number of image sequence is insufficient to support such a strong claim about composition.
* Overall, the experimental results seem a little weak.
  * The model is outperformed on most tasks in the video-prediction setting. It is unclear why the one of the weakest baselines was used for the for action-conditioning comparison in section 4.1.
  * In the visual planning experiment, the performance exceeds baselines in only 2/7 tasks.
  * In the model-based RL experiments, the performance is mostly comparable to dreamer. There are cases where the paper argues that their suggested algorithm outperforms dreamer but given that the results are only reported over $5$ random trials, it seems that the results are within variance making this claim too vague. (It’s a little hard to tell because shaded regions are very transparent)
* The benefits of scalability for the downstream control tasks have not been demonstrated.
  * The work argues that high perceptual metrics don't necessarily correlate with good control performance (App B.2) but the main benefit of scale seems to be improved performance on those metrics.
  * While the work shows in section 4.4 that scaling leads to lower validation loss, whether or not this is correlated with downstream performance is not demonstrated. I believe this point is important because we have to ask what is the marginal benefit of increasing computational complexity to train larger and larger models, if the application to downstream tasks does not benefit much from scale. Showing that larger networks and lower error lead to improved downstream performance is an experiment that might validate the need for scale.
* Similarly, the need for specialized tokenization has not been argued for downstream applications. It seems a little detached from the goal. The main goal of the paper is to argue for scalability and that this tokenization technique is helpful. However, for the downstream applications, neither the planning nor RL sections demonstrate the importance of scaling or tokenizing. One experiment I can think of is to demonstrate that 4x4 tokenization is insufficient due to reconstruction quality this claim. The relationship between scaling and the novel tokenization is not analyzed either.
Overall, I think the motivation of the work is not quite clear enough, the contributed algorithm provides minor improvements in the control settings and the claims for interactivity and scalability could be strengthened.
* The text claims that the tokenization that is presented is more efficient but there is no experiment validating that this is true. A complex encoder structure is used and it is unclear whether this is actually more efficient. An experiment to validate this could measure total wallclock times on the same hardware.

[1] Mamba: Linear-Time Sequence Modeling with Selective State Spaces. Albert Gu, Tri Dao. arXiv preprint arXiv:2312.00752.
[2] Action-Conditional Video Prediction using Deep Networks in Atari Games. Junhyuk Oh, Xiaoxiao Guo, Honglak Lee, Richard L. Lewis, Satinder Singh. Advances in Neural Information Processing Systems 28 (NIPS 2015).

**Questions:**

Q1: What are the training times of your algorithm training and how do they compare to the runtimes of Dreamer?
Q2: Can you elaborate on the term interactive and why other models are not interactive?
Q3: What are the key differences between the proposed model and the MaskVIT model and to which of the difference do you attributed the improved action-conditioned prediction performance in section 4.1?

**Limitations:**

The paper contains an explicit section outlining limitations of the work.

---

> ### Author Rebuttal · Authors · 2024-08-07
>
> We sincerely thank Reviewer 6ajb for providing a detailed review and insightful questions. We have **responded to common questions in the global response and individual questions below**.
>
> ### 1. Motivation of iVideoGPT: Best of both interactivity and scalability
>
> In $\underline{\text{Q1 of global response}}$, we have clarified that iVideoGPT is motivated by two lines of research and aims to achieve both interactivity and scalability.
>
> To specifically answer Reviewer 6ajb's question:
>
> - Interactivity: We define the term "interactive" as that a model can predict future steps incrementally, conditioned on intermediate actions (as defined in Sec 2 of paper). While **Dreamer and many classic models do have interactivity**, we remark that **many advanced video generation models do not**. These include well-known Sora and action-conditioned autoregressive models like VideoGPT as shown in Fig 2b of paper.
>
> - Scalability:
>     - Dreamer uses an RNN architecture which has weaker scalability (**defined and discussed with evidence in global response**).
>     - Why scalability is important? Pre-training foundation models with large-scale data is a well-established paradigm in many other fields of deep learning (e.g. languages and the closely related field of video generation), providing impressive success, but is **under-explored and not yet commonly applied in model-based control**.
>     - While the reviewer mentioned Mamba, we argue that Mamba's macro architecture is highly akin to Transformers, with interleaved temporal modules (SSM) and temporal-independent MLPs, which is far from Dreamer. To our knowledge, **Mamba has not been used in world models and no such Dreamer variants exist in the literature**.
>
> ### 2. Tokenization
>
> ***Context-dynamics decomposition***
>
> To support the decomposition property of the tokenization, we visualized the decoder's reconstruction in $\underline{\text{Fig 1 of attached PDF}}$ by removing cross-attention for future frames. We observed that, without cross-attention, the decoder still reconstructs a trajectory **moving in the same way** as the original, but the **context is almost entirely missing**.
>
> ***Computational efficiency***
>
> We measure time and memory usage on the same hardware, which is reported in $\underline{\text{Q2 of global response}}$. Despite the complexity of our encoder-decoder structure, it only takes up a little more resources than a vanilla 16x16 per-frame tokenizer and saves a lot during training and generation of the transformer.
>
> ***Relationship with scaling***
>
> Efficient tokenization greatly saves computational comsuption for training and generation (rollouts), allowing us to **scale up the model size with fewer costs**.
>
> ### 3. Experiments
>
> Our experiments aim to demonstrate that **one architecture and pre-trained model can be adapted into various downstream tasks to achieve competitive performance**. Although we do not significantly outperform state-of-the-art methods on all tasks, we believe it is a valuable step.
>
> ***Video prediction baselines***
>
> We only compare with MaskViT on action-conditioned BAIR and high-resolution RoboNet settings, since all other models do not report these settings and many strong baselines such as FitVid or MAGVIT do not release official training codes or instructions.
>
> ***Visual planning performance***
>
> We find that mixed results across models in the VP2 benchmark can be primarily attributed to its imperfect built-in reward design. Please refer to Q4 in global response for details.
>
> ***Significance of MBRL experiments***
>
> To better show the statistical significance of our superior performance, we follow the protocol from [1] and report aggregated performance using Inter-Quantile Mean across all 30 runs in $\underline{\text{Fig 5 of attached PDF}}$. It shows that **our method significantly outperforms DreamerV3**, a well-tuned strong baseline for this setting.
>
> [1] Deep reinforcement learning at the edge of the statistical precipice
>
> ***Control benefits of scalability***
>
> There are two kinds of scaling: **model scaling and data scaling**. In our experiments, we currently do not find that further model scaling can significantly improve performance on Metaworld tasks (iVideoGPT-436M performs on par with 138M), likely because Metaworld is a visually simple simulation environment. However, as shown in Fig. 6 of the paper, **data scaling does provide benefits**---pretraining iVideoGPT enhances MBRL performance. In contrast, Dreamer's lack of scalability limits its potential to benefit from pre-training (see Figs 2 \& 5 in attached PDF).
>
> ### 4. Others
>
> ***MBRL training time***
>
> Our method in PyTorch takes ~4 hours on 4090 to train for 100k environment steps. Official DreamerV3 in JAX takes ~1 hour. When fairly compared under the same framework, a DreamerV3 implementation in PyTorch takes ~3.5 hours for 100k steps, comparable to ours.
>
> ***Difference with MaskViT***
>
> iVideoGPT and MaskViT are **fundamentally different types of generative models**. MaskViT is a masked model that generates all video frames simultaneously through masked reconstruction, typically trained for fixed-length videos. In contrast, iVideoGPT is an autoregressive model, allowing flexible video generation of various lengths. Additionally, as discussed in $\underline{\text{Sec 4.1 of paper}}$, MaskViT uses per-frame tokenization and suffers from temporal inconsistencies, while iVideoGPT employs a novel **tokenization conditioned on consistent contextual information**, contributing to its improved performance.
>
> ***Clarification on claims***
>
> We apologize for the strong statements and will revise them for accuracy:
>
> - **"broad world knowledge"** => "common knowledge of motions and interactions in various scenes through pre-training on diverse human and robotic manipulation videos."
> - **"decouples dynamics information from context conditions"** => "designed to encourage the decoupling of dynamics information from context conditions."

---

> > ### Comment · Reviewer_6ajb · 2024-08-13
> >
> > Dear authors, thank you for the thorough response. My initial questions have been answered satisfactory. I also appreciate all the efforts you put into additional experiments.
> >
> > I have one follow-up question since I'm not too familiar with vision tokenizers for video prediction.
> >
> > Q5: In the paper, the text states "Instead of [...]  using a 3D tokenizer that compresses videos spatiotemporally at the expense of interactivity, we propose to tokenize videos with a novel conditional VQGAN"
> > Could you elaborate on why we cannot simply use the VQVAE encoder from the original VideoGPT and why that loses interactivity? What exactly distinguishes this tokenizer from VQVAE?

---

> > > ### Author Response · Authors · 2024-08-13
> > > **Response to Post-rebuttal Feedback by Reviewer 6ajb**
> > >
> > > Dear Reviewer 6ajb,
> > >
> > > Thank you again for your time and effort in reviewing our paper. We appreciate your careful review of our rebuttal materials and your recognition of our efforts in addressing your initial concerns.
> > >
> > > Regarding your follow-up question: What is the difference between the tokenizer in VideoGPT and ours, and how do they impact interactivity?
> > >
> > > VideoGPT uses a VQVAE for video that relies on **a series of 3D convolutions to downsample across space and time**. For example, it downsamples original pixels from $16 \times 64 \times 64$ to discrete tokens of $8 \times 32 \times 32$ or $4 \times 16 \times 16$ (depending on the downsampling ratio). The key issue is that this non-causal downsampling over the temporal dimension results in each token containing information from a window of frames. As a result, **the entire video can only be reconstructed after VideoGPT generates all tokens**. As shown in Fig. 2b of our paper, **VideoGPT only allows the input of future action sequences at the beginning of prediction, preventing an agent from interactively determining its actions based on predicted observations**. In contrast, our tokenizer discretizes video frames separately, using a conditional mechanism to handle temporal redundancy, **enabling frame-by-frame video generation and allowing for intermediate action intervention**.
> > >
> > > Moreover, our tokenizer’s novel design, with its cross-attention mechanism, is more efficient in handling temporal redundancy, converting videos into significantly fewer tokens ($L=511$ with $N=256, n=16, T=16, T_0=1$ as stated in Line 124). In contrast, VideoGPT finds that using a larger downsampling ratio than a token size $8 \times 32 \times 32$, results in worse performance.
> > >
> > > We hope this response addresses your remaining concerns, and we remain open to further discussion. If our response is satisfactory, we kindly ask you to consider re-evaluating your rating of our work based on the clarified understanding.
> > >
> > > Best regards,
> > >
> > > Authors

---

> > > > ### Comment · Reviewer_6ajb · 2024-08-13
> > > >
> > > > Dear authors, thank you for the follow-up response. I will use this comment to summarize my position now that I believe I have a better understanding of the manuscript.
> > > >
> > > > **Motivation**
> > > > First, I appreciate you rephrasing the hyperbolic language. It makes the claims sound much more precise and fosters clarity of what is provided.
> > > >
> > > > That being said, I think that the scalability and interactivity arguments are not fleshed our sufficiently. For instance, [1 mine] also have an interactive world model that is scalable which is probably worth citing. What differentiates the presented work here is that the world-model operates in pixel-observation space. This in itself is interesting because I am not aware of any work that has done this on this complexity. The second argument that the rebuttal makes is that **one** model for downstream fine-tuning is what is being provided. However, this point does currently not play a central role in the writing.
> > > >
> > > > Overall, I think the paper would benefit strongly from re-focusing the main story on pixel-observation spaces and a single-model for downstream learning rather than the interactive aspect. I believe the interactive and scalable aspects are not necessarily what makes this work stand out (I personally would adjust the title to include pixel-spaces/observation spaces. Note that that does not mean the name of the model needs to change since interactive video model is still true.).  A single, pixel observation-based world model for easy downstream finetuning would fit the story of this work better.
> > > >
> > > > **Tokenizer**
> > > > I appreciate all the clarifications on this. Now that I understand better, I believe the text again would benefit from re-writing, focusing on the need of this tokenizer to enable embedding actions seemlessly while also providing an efficient encoding similar to what is needed for video prediction in iVideoGPT rather than things like context-lengths. This would tie the introduction of the tokenizer much better to the objective of building a world model which is currently missing.
> > > >
> > > > **Experiments**
> > > > Thank you for running the additional experiments. I think the experiment section of this paper is going to be quite extensive which is great. Overall, I believe my point about the strengths of the experiments remains. Yet, given my previous explanation we have to put them in a different light. A world model in pixel observation space that is equally good as latent space models is a decent contribution. Tying the language of the claims to the experiments is what remains.
> > > >
> > > > One crucial addition are the additional dreamer experiments to strengthen the claim that previous models are not scalable. However, as I pointed out, Dreamer is not the model that claims to be scalable, but rather [1] does. I understand that there is no time in the rebuttal to consider a comparison here but I do think that comparison would make this paper a stronger submission. Since there is no time for this now, I will not include this into my recommendation to the author's disadvantage but rather provide this as feedback for the next iteration of the paper.
> > > >
> > > > "However, as shown in Fig. 6 of the paper, data scaling does provide benefits---pretraining iVideoGPT enhances MBRL performance."
> > > > Figure 6 only demonstrates that pretraining is useful which is something we have known from offline RL, not that we need a lot of fine-tuning data. There are no experiments with varying amounts of data for **down-stream** performance. In Figure 8b the paper demonstrates that validation loss of pre-training goes down with larger models but in the rebuttal it's stated that that has no impact on the down-stream performance on meta-world. Thus, I believe there is still a disconnect between analyzing the scalability (both model and data) and down-stream performance of the model.
> > > >
> > > > Thank you also for clarifying training times, I underestimated the reduction in computational cost from your tokenizer.
> > > >
> > > > **Summary**
> > > > Overall, I think this paper makes a decent contribution and others might benefit from it being published. However, I think there are various sections that I personally would recommend re-writing. If this was a journal paper, it would be an accept with revisions. If we had more time, I would recommend borderline acceptance and raise my score to 5. Unfortunately, we don't have this option here which puts us at an impasse. Thus I am going to err on the side of optimism and trust that the authors incorporate large parts of my feedback and I will recommend acceptance. Most of the issues that I have with the paper can be fixed by (in some parts minor) re-writing. I am changing my scores as follows:
> > > >
> > > > Presentation: 3 -> 2
> > > > Contribution: 2 -> 4
> > > > Overall score: 3 -> 6
> > > >
> > > > [1] TD-MPC2: Scalable, Robust World Models for Continuous Control. Nicklas Hansen, Hao Su, Xiaolong Wang. ICLR 2024.

---

> > > > > ### Author Response · Authors · 2024-08-13
> > > > > **Deep Appreciation for Your Constructive Feedback**
> > > > >
> > > > > Dear Reviewer 6ajb,
> > > > >
> > > > > We deeply appreciate your time and effort in providing such detailed and constructive feedback. Your recognition of our contributions is greatly encouraging. Your insights have been invaluable in helping us strengthen the clarity and focus of our paper. We are committed to incorporating your feedback and revising the relevant parts on motivation, tokenizers, and experiments to ensure the paper meets your expectations and reaches its full potential.
> > > > >
> > > > > Thank you again for your support and for recommending our work for acceptance.
> > > > >
> > > > > Best regards,
> > > > >
> > > > > Authors

---

> ### Comment · Area_Chair_UXfq · 2024-08-13
> **Required Action: Please Respond to the Author Rebuttal**
>
> Dear Reviewer 6ajb,
>
> As the Area Chair for NeurIPS 2024, I am writing to kindly request your attention to the authors' rebuttal for the paper you reviewed.
>
> The authors have provided additional information and clarifications in response to the concerns raised in your initial review. Your insights and expertise are invaluable to our decision-making process, and we would greatly appreciate your assessment of whether the authors' rebuttal adequately addresses your questions or concerns.
>
> Please review the rebuttal and provide feedback. Your continued engagement ensures a fair and thorough review process.
>
> Thank you for your time and dedication to NeurIPS 2024.
>
> Best regards,
>
> Area Chair, NeurIPS 2024

---

### Official Review · Reviewer_yh7R · 2024-07-11

**Soundness:** 4
**Presentation:** 4
**Contribution:** 4
**Rating:** 7
**Confidence:** 3

**Summary:**

This paper introduces Interactive VideoGPT (iVideoGPT), which builds world model based on VideoGPT architecture. iVideoGPT proposes compressive tokenization, and the model is trained using millions of human or robot videos (i.e., Open X embodiment dataset). The effectiveness of iVideoGPT is demonstrated in video prediction, visual planning, and visual model-based reinforcement learning.

**Strengths:**

- The empirical validation is extensive including video prediction, visual planning and visual model-based reinforcement learning.
- The scalability of iVideoGPT is impressive.

**Weaknesses:**

- Technical novelty is somewhat limited.

**Questions:**

- What is main difference between iVideoGPT and  IRIS [1] for visual model-based reinforcement learning of visual planning?

[1] Micheli, V., Alonso, E., & Fleuret, F. (2022). Transformers are sample-efficient world models. arXiv preprint arXiv:2209.00588.

**Limitations:**

See Weakness & questions.

---

> ### Author Rebuttal · Authors · 2024-08-07
>
> We sincerely appreciate Reviewer yh7R for providing a thorough review, valuable questions, and a positive evaluation of our paper.
>
> ### Q1: Technical novelty
>
> Discrete tokenization and autoregressive transformer are prevelant in contemporary deep learning, due to their simplicity and generality. iVideoGPT generally shares this architecture with IRIS, but possesses distinguishing features (discussed below in Q2).
>
> Despite adopting a widely used architecture, we believe the most important contribution of our paper lies in **advancing a paradigm of pre-training a scalable world model architecture on large-scale data and adapting it into various downstream tasks**. This paradigm is well-established in many other fields of deep learning (e.g., languages and the closely related field of video generation) but is not yet commonly applied in MBRL (see $\underline{\text{Q1 of global response}}$ for extended discussion).
>
> ### Q2: Difference between iVideoGPT and IRIS
>
> We summarize four key features of iVideoGPT that are different from IRIS:
>
> 1. Pre-training and fine-tuning paradigm enhancing sample efficiency;
> 2. Novel and efficient tokenization enabling **faster rollouts** (**~24x**, see Q2 in global response);
> 3. Flexible action-conditioning design;
> 4. Off-policy MBRL implementation.
>
> All of these features are highly relevant to visual planning and visual MBRL. Please refer to $\underline{\text{Q3 of global response}}$ for a more detailed discussion.

---

> ### Comment · Area_Chair_UXfq · 2024-08-13
> **Required Action: Please Respond to the Author Rebuttal**
>
> Dear Reviewer yh7R,
>
>
> As the Area Chair for NeurIPS 2024, I am writing to kindly request your attention to the authors' rebuttal for the paper you reviewed.
>
> The authors have provided additional information and clarifications in response to the concerns raised in your initial review. Your insights and expertise are invaluable to our decision-making process, and we would greatly appreciate your assessment of whether the authors' rebuttal adequately addresses your questions or concerns.
>
> Please review the rebuttal and provide feedback. Your continued engagement ensures a fair and thorough review process.
>
> Thank you for your time and dedication to NeurIPS 2024.
>
>
> Best regards,
>
> Area Chair, NeurIPS 2024

---

### Official Review · Reviewer_zSnp · 2024-07-12

**Soundness:** 2
**Presentation:** 3
**Contribution:** 3
**Rating:** 6
**Confidence:** 4

**Summary:**

This paper introduces a new architecture for an action-conditioned video (and reward) prediction model.
First, a VQGAN converts context frames individually into tokens.
Second, a conditional VQGAN converts future frames individually into tokens, conditioned on the context frames (using intermediate representations from the context encoder).
The idea is that the future encoder can focus on encoding changes in the scene, since the remaining information was already extracted from the context, allowing for a lower number of tokens.
Then, an autoregressive transformer is trained to predict the next (future) token and is optionally conditioned on actions and can optionally predict the rewards in a reinforcement learning setting.
The model is pre-trained on action-free robotic manipulation videos and then fine-tuned and evaluated on video prediction, visual planning, and model-based reinforcement learning tasks.

**Strengths:**

- S1: Encoding the dynamics information conditioned on context information is an intriguing idea for world models.
- S2: The paper is clearly written and the visualizations are informative and appealing.

**Weaknesses:**

- W1: The paper is missing quantitative comparisons of the computational efficiency. For instance, it would be interesting to see the differences in training/inference time when using the different tokenizers (Figure 8(c)).
      Furthermore, the authors state at the end of Section 2 that recurrent world models like Dreamer are not scalable, but I'm wondering how significant the difference is, considering the autoregressive next token prediction and use of multiple tokens per frame.
- W2: The authors argue that iVideoGPT eliminates the need for latent imagination in model-based RL (Section 4.3 and Figure 5).
      However, there are existing world models (e.g. IRIS [1]) that learn behaviors using reconstructed frames.
      Moreover, Dreamer could also learn behaviors this way by using the reconstructions from the decoder (but I understand that this would be a different method).
      In short, I don't think that this is a feature that is novel or specific to iVideoGPT.
- W3: The proposed model has a lot in common with IRIS [1], which learns a discrete autoencoder (VQVAE) and uses an autoregressive transformer for next token prediction.
      IRIS is briefly mentioned in the related work section, but I think the authors could emphasize the differences in more detail.

Some typos:
- 79: "aims"
- 80: "maximize"?
- 85: "history of $T_0$ video frames"?
- 86: "needs to"
- 107: $D_c(z_t)$ should be $D_c(z_t^{(1:N)})$?
- Also in Eq. (1) it should be $D_p(z_t^{(1:n)}|o_{1:T_0})$?
- 687: Table 2 -> Table 3

I am willing to increase my scores after the listed weaknesses have been addressed.

**Questions:**

- Q1: Is the world model also conditioned on the actions from the context? In line 87 the condition only includes $a_{T_0:t}$, but in the sequence of tokens $x$, the context also includes slot tokens. This can also not be recognized in Figure 3(b).

**Limitations:**

The authors addressed all limitations adequately.

---

> ### Author Rebuttal · Authors · 2024-08-07
>
> We sincerely thank Reviewer zSnp for the thorough review and valuable questions. We also appreciate the pointed-out typos, which we will correct in a future revision.
>
> ### W1: Computational efficiency
>
> We apologize for missing a quantitative efficiency analysis. We have reported training/inference time and memory usages with various tokenizers, in $\underline{\text{Q2 of global response}}$. Our proposed compressive tokenization provides **significant memory savings during training and faster rollouts during generation**.
>
> #### Discussion on Dreamer
>
> While we have enhanced the scalability of Transformer-based world models through compact tokenization and resource savings, Dreamer with an RNN architecture may still be more efficient computationally. However, as demonstrated in $\underline{\text{Fig 2 of attached PDF}}$, when pre-training Dreamer XL (200M parameters, comparable to iVideoGPT) on the same dataset as us, we find that it has **insufficient capacity to support large-scale pre-training**, which is crucial for the success of modern foundation models. The results in $\underline{\text{Fig 5 of attached PDF}}$ further prove that Dreamer is **unable to benefit from such ineffective pre-training**.
>
> ### W2: Eliminating latent imagination
>
> We do not claim that eliminating latent imagination is a unique feature of iVideoGPT. However, since latent imagination is currently the dominant practice in MBRL (as seen in Dreamer, MuZero, etc.), we believe it is an advantage for iVideoGPT to simply serve as a plug-in replacement of the environment. As discussed in Sec 4.3, this can simplify the design space, reduce implementation complexity, and enhance the practicality of MBRL.
>
> The key to this advantage is a powerful world model. Previous action-conditioned video prediction models like FitVid, without sufficient capacity and pretrained knowledge, can also function similarly to iVideoGPT but can **produce more blurring predictions** ($\underline{\text{Fig 4 of attached PDF}}$), thus **hinders MBRL directly on top of these inaccurate frames** ($\underline{\text{Fig 5 of attached PDF}}$). These results may explain why Dreamer employs latent imagination in MBRL. While IRIS shares a similar architecture with iVideoGPT, it lacks efficient tokenization and large-scale pre-training (as discussed below).
>
> ### W3: Difference with IRIS
>
> We summarize four key differences between our approach and IRIS. Please refer to $\underline{\text{Q3 of global response}}$.
>
> ### Q1: Conditioning on the context actions
>
> In our implementation, iVideoGPT is not conditioned on context actions. However, it can easily be extended to support this by adding the embedding of context actions to the slot tokens between context frames. We hypothesize that this would not significantly impact performance, as context actions can likely be inferred implicitly from context frames.

---

> ### Comment · Area_Chair_UXfq · 2024-08-13
> **Required Action: Please Respond to the Author Rebuttal**
>
> Dear Reviewer zSnp,
>
>
> As the Area Chair for NeurIPS 2024, I am writing to kindly request your attention to the authors' rebuttal for the paper you reviewed.
>
> The authors have provided additional information and clarifications in response to the concerns raised in your initial review. Your insights and expertise are invaluable to our decision-making process, and we would greatly appreciate your assessment of whether the authors' rebuttal adequately addresses your questions or concerns.
>
> Please review the rebuttal and provide feedback. Your continued engagement ensures a fair and thorough review process.
>
> Thank you for your time and dedication to NeurIPS 2024.
>
>
> Best regards,
>
> Area Chair, NeurIPS 2024

---

> > ### Comment · Reviewer_zSnp · 2024-08-13
> >
> > Thank you for the detailed response. I have updated my score accordingly.

---

> > > ### Author Response · Authors · 2024-08-13
> > > **Appreciation for Your Support**
> > >
> > > Dear Reviewer zSnp,
> > >
> > > We sincerely appreciate your dedicated re-evaluation of our paper and the subsequent positive rating. Your feedback has significantly improved our work.
> > >
> > > Best regards,
> > >
> > > Authors

---

### Official Review · Reviewer_HkK9 · 2024-07-12

**Soundness:** 3
**Presentation:** 3
**Contribution:** 2
**Rating:** 5
**Confidence:** 4

**Summary:**

The authors present a paper that attempts to utilize actionless and action conditioned trajectories to learn a large scale interactive world model. This model is subsequently adapted for robot manipulation tasks. It is evaluated on video prediction, visual planning, model based RL. The model training is tested primarily on Metaworld in the embodied setup. The experiments demonstrate the generalization capabilities of the model.

**Strengths:**

1. The paper is well written, clear and easy to understand
2. The idea is well motivated. The question of utilizing internet scale knowledge for embodied intelligence is an open area of research.
3. The experiment results are statistically significant with multiple runs reported accompanied by error bars
4. The authors present a novel tokenize scheme which can benefit other video foundation models

**Weaknesses:**

1. **Performance on Video prediction:** The performance for the visual planning is rather unsatisfactory. Do the authors have any intuition about why their method only outperforms the baselines in 2 of the setups?
2. **No analysis on amount of action data needed:** During the motivation of the method, the authors discuss coming up with methods that are able to learn from freely available videos. Robot data on the other hand is expensive. Thus, ablations on how much robot data is needed are necessary to understand the dependence of the model on robot specific data and whether or not the method is indeed benefitting from freely available human videos.
3. **Lack of robot experiments:** Does the model extend to any real robot setups? Currently, the only low level control robot experiments available are on 6 in which the model matches the performance in 3.
4. **No human user studies:** the paper only uses numerically metrics like Frechet Video Distance to judge the quality of generation. This metric does not always align with quality. The results of such a study would help bolster the quality of the work.
5. **Some missing Baselines**: FitVid does indeed have an action conditioned model and this model could be used for all the MBRL experiments. This currently missing.

**Questions:**

1. Is the dreamer baseline in the paper trained on OpenX data also? If not then the comparison is not fair.
2. How do you ensure that the $z^(1:n)_t$ contain all the information about future frames and that there is no leakage between which tokens contain what information?

**Limitations:**

1. The evaluation suite is currently very small should be expanded
2. Its unclear how much the human videos from Something-something-v2 help the method. If they do not help the method significantly, then this diminishes the contributions of the paper significantly as this would make it a study on the application of transformer based world modeling on large robot datasets like OXE.

Inspite of these limitations and weaknesses, I think the work makes interesting contributions and I would inclined to increase my score if the authors are able to address my questions.

---

> ### Author Rebuttal · Authors · 2024-08-07
>
> We would like to sincerely appreciate Reviewer HkK9 for the comprehensive review and insightful questions. We have **addressed common questions in the global response and indivual questions below**.
>
> ### Q1: Amount of action data needed
>
> As described in $\underline{\text{Sec 3.2}}$, we **do not use any action data during pre-training**. Leveraging more accessible, action-free video data reduces the cost of data collection. Although action-free robot videos are still more expensive than Internet videos, there has been significant progress in data sharing (e.g., BridgeData, RH20T, OXE), making robot data increasingly affordable. **What we truly need are unified frameworks like iVideoGPT to better utilize these**.
>
> **Expensive action-conditioned data are only used for downstream tasks**. In $\underline{\text{Sec 4.4}}$, we have demonstrated the few-shot adaptation capabilities for both (action-free) video prediction (Fig 7, 8c) and MBRL (Fig 6). In addition, we have also adapted iVideoGPT with 1,000 action-conditioned BAIR trajectories, which achieves 82.3 FVD.
>
> ### Q2: Contribution of human videos
>
> As detailed in Appendix A.2, we used a mixture of OXE and SthSth for pre-training the tokenizer, as visual diversity is the key to learning context and dynamics information separation. The transformer is pre-trained on OXE only, as we found many SthSth videos with random camera motions are hard to predict. These choices were based on our experience and a comprehensive analysis is expensive and left for future work.
>
> To assess the contribution of human videos, we pretrain iVideoGPT with pure OXE data (taking almost a week) and evaluate on OXE validation data. We observe human videos indeed help pre-training.
>
> | Pretrain data | FVD      | PSNR     | SSIM     | LPIPS   |
> | ------------- | -------- | -------- | -------- | ------- |
> | OXE only      | 48.3     | 24.3     | 85.5     | 9.1     |
> | OXE+SthSth    | **40.5** | **24.8** | **86.3** | **8.5** |
>
>
> ### Q3: MBRL baseline with FitVid
>
> Although FitVid is originally designed for video prediction and has not been used as world model for MBRL, we have implemented a baseline using FitVid, by replacing iVideoGPT in our implementation. To predict rewards, we add an MLP head on top of FitVid's latent state, parallel to the image decoder.
>
> As shown in $\underline{\text{Fig 5 of attached PDF}}$,  MBPO with iVideoGPT outperforms FitVid on 5/6 tasks and performs comparably on the remaining one. We also qualitatively observe that FitVid's imagined trajectories are blurrier (as shown in $\underline{\text{Fig 4 of attached PDF}}$) compared to ours, which hinders its ability to accurately simulate real environments and may hurt MBRL performance.
>
> ### Q4: Dreamer baseline with pre-training
>
> As elaborated in $\underline{\text{Q1 of global response}}$, Dreamer, with its RNN architecture, lacks the scalability to benefit from pre-training on real-world videos [1]. To empirically validate this, we have used a pre-training algorithm, APV [1], to pre-train DreamerV3-XL (200M parameters, comparable to iVideoGPT) on the same pre-training data as ours, and then finetune it on Metaworld tasks.
>
> The results in $\underline{\text{Fig 5 of attached PDF}}$, indicate that Dreamer almost does not benefit from pre-training. We also visualize DreamerV3's predictions after pre-training in $\underline{\text{Fig 2 of attached PDF}}$, which is of greatly lower quality compared to iVideoGPT, validating the limited capacity of Dreamer architecture.
>
> [1] Reinforcement Learning with Action-Free Pre-Training. ICML 2022.
>
> ### Q5: Human study
>
> As official pretrained models are not released for most baselines, we are only able to compare iVideoGPT with VideoGPT and MCVD on the action-free BAIR dataset. We generate videos using three models from the test set and ask users to label preferences between two randomly sampled videos, based on the physical naturalness and feasibility of robot-object interactions.
>
> We collect 386 annotations from 9 persons. The results in $\underline{\text{Fig 6 of attached PDF}}$ demonstrate that iVideoGPT is more preferred by human annotators.
>
> ### Q6: Lack of real-robot experiments
>
> We would like to clarify that, instead of being 'tested primarily on Metaworld in the embodied setup', our experiments in Sec. 4.2 (visual planning) and Sec. 4.3 (visual MBRL), although conducted in simulation, are **both embodied low-level control settings**.
>
> We apologize that the limited time of rebuttal is insufficient for us to set up a real-robot experiment. We do not claim a contribution that our method is instantly ready for real-robot applications, which is left for future work, but we believe our method can help advance model-based methods in robotics research.
>
> ### Q7: Visual planning performance
>
> We find that inconsistent performance in the VP2 benchmark can be primarily attributed to inaccurate built-in reward design of this benchmark. Please refer to Q4 in global response for details.
>
> ### Q8: "Small" evaluation suite
>
> We respectfully disagree with "The evaluation suite is currently very small". In fact, **no baselines cover all three problem settings in our paper**. We hope additional results provided in this rebuttal can solidify our evaluations.
>
> ### Q9: Context-dynamics decomposition in tokenization
>
> While we cannot guarantee in neural networks perfect decomposition and no information leakage, we aim to achieve this through a careful tokenizer design: a bottleneck with much fewer tokens compels to only capture necessary dynamics information for future frames and share contextual information with initial frames, to reconstruct raw pixels.
>
> We visualize the decomposition property in $\underline{\text{Fig 1 in attached PDF}}$. The showcases illustrate that, by removing cross-attention from context frames into future frames, the decoder can still reconstruct a trajectory **moving the same way** as the normally reconstructed one but the **context is almost missing**.

---

> ### Comment · Area_Chair_UXfq · 2024-08-13
> **Required Action: Please Respond to the Author Rebuttal**
>
> Dear Reviewer HkK9,
>
>
> As the Area Chair for NeurIPS 2024, I am writing to kindly request your attention to the authors' rebuttal for the paper you reviewed.
>
> The authors have provided additional information and clarifications in response to the concerns raised in your initial review. Your insights and expertise are invaluable to our decision-making process, and we would greatly appreciate your assessment of whether the authors' rebuttal adequately addresses your questions or concerns.
>
> Please review the rebuttal and provide feedback. Your continued engagement ensures a fair and thorough review process.
>
> Thank you for your time and dedication to NeurIPS 2024.
>
>
> Best regards,
>
> Area Chair, NeurIPS 2024

---

### Author Rebuttal · Authors · 2024-08-07

We sincerely thank all reviewers for their constructive feedback. We have made every effort to address all concerns and have responded to individual reviews. We have also **answered common questions in this global response**.

Please note that **all the new figures for all responses are included in the PDF attachment**.

### Q1: Motivation of iVideoGPT

As discussed in Sec. 1 of the paper, iVideoGPT is **motivated by two distinct lines of research**:

 - **World models** in RL serves as an **interactive** simulation of the real environments by modeling transitions. However, the dominant architectures are still based on RNNs (like Dreamer and MuZero), which have **limited scalability** compared to Transformers (see Fig 7 in [1]; here **scalability means an ability to increase capacity effectively with additional parameters**). Thus, current world models rarely do pre-training on large-scale real-world videos and struggle to benefit from it (see Fig 8c in [2] and Figs 2 \& 5 in attached PDF), leading to insufficient sample efficiency in model-based control.

 - **Video generation models** (e.g. Sora) are recently advanced by **scalable** architecture and large-scale training. They share advantages with foundation models in other fields (like LLMs): strong performance with sufficient task data, fast adaptation with few-shot downstream samples, and zero-shot generalization on unseen domains. However, these models typically generate all video frames simultaneously, allowing text/action conditions only at the start, thus **lacking the ability for interaction** during generation.

To bridge both sides, our work is a framework where **scalable and interactive architecture can be pre-trained with large-scale data and then adapted into various downstream tasks**. Across three control-relevant settings, iVideoGPT demonstrates favorable properties, including **competitive task performance** (Tab 1, Figs 4\&6 in paper), **few-shot adaptation** (Fig 8a in paper), and **zero-shot generalization** (Fig 7 in paper).

While many future directions remain to be explored, we believe that our work as a valuable prototype towards large world models, will contribute to the community.

[1] Scaling laws for neural language models. 2020.
[2] Reinforcement Learning with Action-Free Pre-Training. ICML 2022.

### Q2: Tokenization Efficiency

In addition to scalable Transformer architecture, we introduce a new compressive tokenization. By exploiting temporal redundancy, a significantly fewer amount of tokens can **greatly save time and memory, allowing us to scale the model size with fewer costs**.

To prove the efficiency, we report the time and memory consumption of iVideoGPT with various tokenizers, showing that our method achieves a good balance between efficiency and quality:

- Pre-training the transformer (A100 with per-device batch size 16)

|Tokenizer|Speed (iters/sec)|Memory (GB)|
|-|-|-|
|4x4|3.10|10.6|
|16x16|N/A|**OOM**|
|Ours|2.62|22.3|

- Inference (RTX 4090 with batch size 1)

|Tokenizer|Tokenize (sec)|Generation (sec)|Detokenize (sec)| Memory (GB)|
|-|-|-|-|-|
|4x4|0.27|1.13|0.05|1.98|
|16x16|0.26|**22.5**|0.04|1.90|
|Ours|0.29|1.11|0.06|2.33|

### Q3: Difference with IRIS

We can summarize the following differences with IRIS:

1. **Pre-training and fine-tuning paradigm**: iVideoGPT is designed for a paradigm that involves pre-training on large-scale videos and fine-tuning on various downstream tasks (see Q1 above). In contrast, IRIS focuses solely on MBRL with Transformer-based world models trained from scratch in the Atari domain.
2. **Efficient tokenization**: iVideoGPT proposes novel compressive tokenization to significantly reduce the number of tokens, saving time and memory (see Q2 above), while IRIS uses per-frame tokenization.
3. **Flexible action-conditioning design**: iVideoGPT employs slot tokens with optional additive action embeddings to support both action-free pre-training and action-conditioned fine-tuning, while IRIS strictly treats discrete Atari actions as tokens.
4. **Off-policy MBRL implementation**: iVideoGPT uses an off-policy RL algorithm while IRIS performs on-policy learning. On-policy learning needs a large number of model rollouts, which, combined with inefficient tokenization, results in ~7 days for 100k-environment-step training. In comparison, iVideoGPT only need takes ~4 hours.

We will include this extended discussion in a future revision.

### Q4: Visual planning performance

We note that **no current model in the VP2 benchmark consistently outperforms other models across all tasks**, and iVideoGPT is no exception. We conducted these visual planning experiments, aiming to show that our model can effectively handle various settings with competitive performance.

When analyzing the inconsistent performance on this benchmark, we primarily attribute it to **imperfect built-in reward designs of VP2**. In this benchmark, scores for sampled actions are mainly estimated by a learned classifier of task success based on model-predicted frames. This classifier, trained by VP2 authors, appears to lack robustness and can be easily fooled by out-of-distribution (OOD) inputs, assigning high rewards to trajectories that are less likely to succeed (see examples in Fig 3 of attached PDF). This imperfect reward function likely contributes to the mixed results observed on this benchmark (with iVideoGPT even outperforming the oracle simulator in one task). Addressing visual planning with imperfect rewards is another independent research problem, out of this paper's scope. We also find our model can produce incorrect predictions for severely OOD actions, likely due to narrow training data. We will include this discussion in the limitation section.

---

### Decision · Program_Chairs · 2024-09-25

**Decision:**

Accept (poster)

**Comment:**

The paper "iVideoGPT: Interactive VideoGPTs are Scalable World Models" presents an approach to world modeling in reinforcement learning using interactive and scalable VideoGPTs. The authors propose a framework where a Transformer-based architecture is pre-trained on large-scale video data and then adapted to various downstream tasks, demonstrating competitive performance, few-shot adaptation, and zero-shot generalization. The reviewers agree that the paper makes a contribution to the field and recommend its acceptance.

The key strengths of the paper lie in its innovative combination of interactivity and scalability in world modeling. By leveraging the advantages of both reinforcement learning world models and advanced video generation models, iVideoGPT bridges the gap between these two domains. The authors introduce a compressive tokenization method that exploits temporal redundancy, which enables the model to scale efficiently while maintaining the ability to generate high-quality videos.

The reviewers raised several important questions and concerns, which the authors addressed comprehensively in their rebuttal. They clarified the motivation behind iVideoGPT, which highlights its potential to benefit from large-scale pre-training and its advantages over existing architectures like RNNs. The authors provided additional experiments and analyses to support their claims, including the efficiency of their tokenization method and the contribution of human videos to pre-training. The authors also addressed the differences between iVideoGPT and related works, such as IRIS and Dreamer, which are two important previous works. They highlighted iVideoGPT's unique features, including its pre-training and fine-tuning paradigm, efficient tokenization, flexible action-conditioning design, and off-policy MBRL implementation. These differences contribute to iVideoGPT's superior performance and sample efficiency compared to existing methods.

While the reviewers acknowledged that iVideoGPT does not significantly outperform state-of-the-art methods on all tasks, they recognized the value of the proposed framework as a step towards large-scale world models. The authors provided additional analyses to explain the mixed results in the visual planning benchmark, attributing them to imperfect built-in reward designs. The authors also responded to concerns about the technical novelty of iVideoGPT. They clarified the differences between iVideoGPT and related works, such as MaskViT, and provided evidence for the effectiveness of their approach in terms of computational efficiency and performance. The reviewers' concerns have been adequately addressed in the rebuttal, and the authors have shown a commitment to improving the paper based on the feedback received.

As a result, I recommend accepting this paper.